# The structure and flexibility analysis of the *Arabidopsis* synaptotagmin 1 reveal the basis of its regulation at membrane contact sites

Juan L Benavente[1], Dritan Siliqi[2], Lourdes Infantes[1], Laura Lagartera[3], Alberto Mills[4], Federico Gago[4], Noemí Ruiz-López[5], Miguel A Botella[5], María J Sánchez-Barrena[1], Armando Albert[1]

**Non-vesicular lipid transfer at ER and plasma membrane (PM) contact sites (CS) is crucial for the maintenance of membrane lipid homeostasis. Extended synaptotagmins (E-Syts) play a central role in this process as they act as molecular tethers of ER and PM and as lipid transfer proteins between these organelles. E-Syts are proteins constitutively anchored to the ER through an N-terminal hydrophobic segment and bind the PM via a variable number of C-terminal C2 domains. Synaptotagmins (SYTs) are the plant orthologous of E-Syts and regulate the ER–PM communication in response to abiotic stress. Combining different structural and biochemical techniques, we demonstrate that the binding of SYT1 to lipids occurs through a Ca²⁺-dependent lipid-binding site and by a site for phosphorylated forms of phosphatidylinositol, thus integrating two different molecular signals in response to stress. In addition, we show that SYT1 displays three highly flexible hinge points that provide conformational freedom to facilitate lipid extraction, protein loading, and subsequent transfer between PM and ER.**

## Introduction

In plants, abiotic stress induces changes in lipid homeostasis that lead to plasma membrane (PM) instability; consequently, plants tend to readjust lipid composition by the action of enzymes involved in lipid modification or by promoting lipid transfer between the membranes of different organelles (Hou et al, 2016; Zhu, 2016). ER–PM contact sites (CS) have an important role in achieving this objective as they have been shown to act as subcellular platforms to integrate intracellular molecular stress signals and non-vesicular lipid transfer (Schapire et al, 2009; Pérez-Sancho et al, 2015; Lee et al, 2019, 2020; Ruiz-Lopez et al, 2020 *Preprint*). CS consist of narrow appositions of plasma and ER membranes, ranging from 10 to 30 nm, which rely on the regulated localization of different molecular tethers and lipid transfer proteins (Fernández-Busnadiego et al, 2015; Krauβ & Haucke, 2016; Wong et al, 2018; Collado et al, 2019).

The molecular players controlling ER–PM connectivity at CS are evolutionarily conserved in all eukaryotic systems and they have developed equivalent biochemical functions in different organisms (Saheki, 2017; Saheki & De Camilli, 2017a). Among them, the extended synaptotagmins (E-Syts) represent a family of conserved ER-localized proteins that display a dual function as molecular tethers between ER and PM and as lipid transfer proteins at CS (Saheki & De Camilli, 2017b; Wong et al, 2018). E-Syts decode the presence of phosphorylated phosphatidylinositol lipids (PIP) at the PM and intracellular levels of calcium ions (Ca²⁺) and, as a result, they enhance their interaction with PM and trigger lipid transport (Saheki et al, 2016; Bian et al, 2018; Collado et al, 2019; Lee et al, 2019). Plant synaptotagmins (*SYTs*) are orthologs of the mammalian E-Syts and trigger an increase in the size and connectivity of CS in response to salt (Lee et al, 2019) and rare earth elements (Lee et al, 2020) or different forms of mechanical stress (Pérez-Sancho et al, 2015).

At the molecular level, E-Syts consist of a hydrophobic segment inserted in the ER followed by a synaptotagmin-like mitochondrial lipid–binding protein (SMP) domain linked to a number of C2 domains (Min et al, 2007; Giordano et al, 2013), which are modules that usually bind to phospholipid membranes in a Ca²⁺-dependent manner (Rizo & Südhof, 1998). Whereas the tethering function of E-Syts lies on the putative hydrophobic hairpin inserted into the ER (Giordano et al, 2013; Pérez-Sancho et al, 2015) and the C2 domains that bind to the PM (Yu et al, 2016), the SMP domain is involved in the specific binding and transport of lipids (Schauder et al, 2014; Saheki et al, 2016; Collado et al, 2019).

[1]Instituto de Química Física "Rocasolano," Consejo Superior de Investigaciones Científicas (CSIC), Madrid, Spain   [2]Istituto di Cristallografia, Consiglio Nazionale delle Ricerche (CNR), Bari, Italy   [3]Instituto de Química Médica (IQM), CSIC, Madrid, Spain   [4]Área de Farmacología, Departamento de Ciencias Biomédicas, Unidad Asociada al IQM-CSIC, Universidad de Alcalá, Madrid, Spain   [5]Departamento de Biología Molecular y Bioquímica. Instituto de Hortofruticultura Subtropical y Mediterránea "La Mayora," Universidad de Málaga-CSIC (IHSM-UMA-CSIC), Universidad de Málaga, Campus de Teatinos, Málaga, Spain

Correspondence: xalbert@iqfr.csic.es

The crystal structure of a fragment comprising the SMP and the first and second C2 domains (C2A and C2B) of human E-Syt2 revealed that the protein dimerizes through the SMP domain forming an elongated structure that defines a long crevice loaded with lipids. In this structure, each C2A–C2B pair forms an independent rigid tandem that displays flexibility with respect to the SMP (Schauder et al, 2014). These data suggested a model in which the SMP dimer would either shuttle or tunnel lipids from the ER to the PM and where the driving force for this movement would be the binding of $Ca^{2+}$ to the C2 domains. However, different additional roles have been assigned to the C2 domains. Among them, the ability to sense cytosolic $Ca^{2+}$ signals to trigger E-Syt binding to the membrane and the specific recognition of phosphatidylinositol-4,5-bisphosphate (PI[4,5]P$_2$)–rich membranes. In addition, a regulatory role in lipid transport has been proposed as it has been seen that they contribute to lipid extraction by inducing membrane curvature (Min et al, 2007; Idevall-Hagren et al, 2015; Pérez-Sancho et al, 2015; Bian et al, 2018; Collado et al, 2019; Lee et al, 2019). Such complexity is encoded on the properties, number and relative position of the C2 domains, and this, in turn, varies among isoforms and organisms. For instance, human E-Syts display either three or five C2 domains (Min et al, 2007); yeast E-Syts, known as tricalbins (Tcbs), either four or five C2 domains (Creutz et al, 2004); and plant E-Syts (SYTs), two domains (Pérez-Sancho et al, 2016).

It is intriguing how plant SYTs achieve the required regulation by using only two C2 domains, given the sophisticated mechanisms proposed for proteins with a higher number of C2 domains. For instance, the five C2 domains of human E-Syt1 provide a regulatory mechanism in which the $Ca^{2+}$-dependent membrane binding of the C2A and C2C domains releases the interaction of the C2A with the SMP domain that inhibits lipid transport and the C2C–C2E interaction that prevents the C2E from interacting with PI(4,5)P$_2$-rich membranes (Idevall-Hagren et al, 2015; Bian et al, 2018). Moreover, the molecular function of Tcbs also involves several C2 domains because the C2AB tandem, together with the N-terminal hydrophobic segment, may be required to generate peaks on the ER facing the PM to facilitate lipid transport, whereas the others may display a PM-tethering function (Collado et al, 2019). Altogether, these data suggest that the plant C2 domains may harbor more specialized and manifold functions than those observed for the E-Syts from other kingdoms. At the same time, SYTs provide a simpler model to study protein activation in response to molecular signals.

To investigate the molecular mechanism responsible for the function of SYTs and specifically the role of the C2 domains in their regulation, we performed structural, biochemical, and computational studies on SYT1. Our data show that the properties of the C2A domain provide the basis for the integration of different signals as it displays independent binding sites to interact with $Ca^{2+}$ and PIPs. We additionally show that SYT1 displays up to three hinge points that confer orientational freedom to the C2 domains and provide flexibility to the molecule. These features enable an intimate contact with the PM upon accumulation of PIP and/or when a $Ca^{2+}$ signal occurs. We propose that this process may generate a tense molecular conformation that results in a shortening of the distance from the SMP domain to the PM to facilitate lipid extraction and loading into the protein.

# Results

## The structure of the C2A domain of SYT1

To frame into a structural model the functions of the C2A domain of SYT1 that differ from those of other C2 domains of the E-Syts, we solved its crystal structure in complex with $Ca^{2+}$ (SYT1C2A, residues 253–397) (Fig 1B and Table S1). Similar to other C2 domains, SYT1C2A folds as a four-stranded $\beta$ sandwich with a $Ca^{2+}$-binding site placed in a cavity formed by three loops connecting the two sheets (L1, L2, and L3) (Fig 1B). The structure of the domain resembles that of the C2A domain of E-Syt2 (E-Syt2C2A), which displays a 25.7% sequence identity and a root mean squared deviation of 1.37 Å for 115 C$\alpha$ atoms (Schauder et al, 2014; Xu et al, 2014). Main differences are found in the loop connecting $\beta$6 and $\beta$7, which is folded as an $\alpha$-helix in E-Syt2C2A and it is not present in SYT1C2A; in the loop connecting $\beta$1 and $\beta$2, which is unusually long in E-Syt1C2A (Xu et al, 2014); and in a long insertion of 12 amino acids connecting $\beta$7 and $\beta$8 (loop L4), which is characteristic of *Arabidopsis* SYT1, SYT2, and SYT3 (Fig 1C). This comparison is relevant to stablish if the unique structural features of SYT1C2A domain are important for SYT1 function.

The core structure of SYT1C2A is similarly found in E-Syt2C2B and E-Syt2C2C domains; however, whereas SYT1C2A, E-Syt2C2A, and E-Syt2C2B display the topology of those C2 domains with the N and C termini opposite to the $Ca^{2+}$-binding site (topology 2), E-Syt2C2C has both ends at the same side of the $Ca^{2+}$-binding site (topology 1) (Fig S1). This molecular organization does not determine the biochemical properties of the individual C2 domains but it might account for the properties of the consecutive C2 domains forming SYT1.

## The mechanism of $Ca^{2+}$-mediated activation of SYT1C2A

Available biochemical data on SYT1 showed that whereas C2B does not harbor any $Ca^{2+}$-binding site, the C2A domain displays $Ca^{2+}$-dependent activity to bind small negatively charged phospholipids such as phosphatidylserine (PS) (Schapire et al, 2008; Pérez-Sancho et al, 2015). To understand the basis of such activity, we analyzed the $Ca^{2+}$-binding properties and the structure of the $Ca^{2+}$-dependent lipid binding site of SYT1C2A.

Our structural data show that SYT1CA2 displays two calcium ions (Ca I and Ca II) which are coordinated by the side chains of Asp 276, Asp 282, Asp 332, Glu 334, and Glu 340 and by the carbonyl oxygen from Lys 275 and Trp 333 (Fig 2A). In addition, two water molecules complete the coordination of $Ca^{2+}$. Lipid binding to this site lies in the direct coordination of the phosphate group of lipids with the $Ca^{2+}$ ions by the replacement of these water molecules (Rizo & Südhof, 1998). E-Syt2C2A site displays a similar structure; however, the site binds three $Ca^{2+}$ ions, with two of them at equivalent positions to those of SYT1C2A and the third, not present in SYT1C2A, bound to L3 (Fig 1B).

To quantify the $Ca^{2+}$-binding properties of SYT1C2A in solution, we used isothermal titration calorimetry (ITC). Our data show that the binding isotherm fits to the one-set-of-sites model in which SYT1C2A binds only one $Ca^{2+}$ (n = 0.221 ± 0.003) with a $K_d$ of 0.06 ± 0.02 $\mu$M ($\Delta$H = −5.3 ± 0.1 kcal•mol$^{-1}$) (Fig 2B). Such nanomolar $Ca^{2+}$-binding site may serve as a structural site stabilizing the C2A domain at

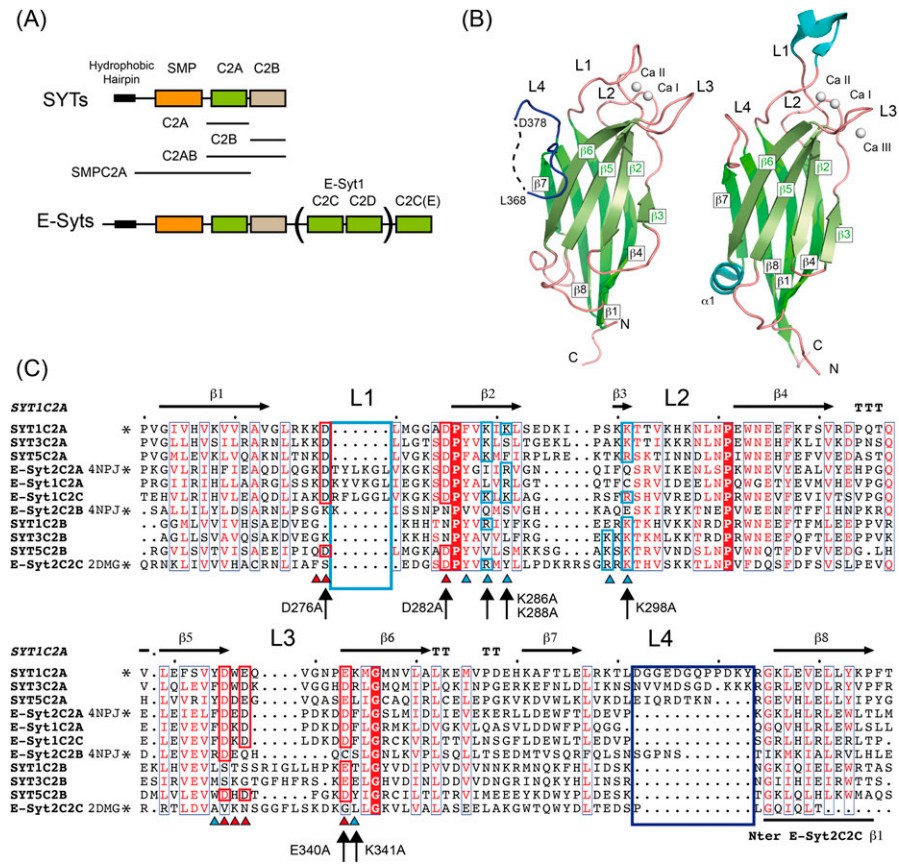

**Figure 1. The structure of the C2A domain of SYT1.**
**(A)** Domain organization of plant (SYTs) and human (E-Syts) extended synaptotagmins. Human E-Syt1 displays five C2 domains, whereas E-Syt2 and E-Syt3 display three. The constructs used in this work are indicated. **(B)** Ribbon representations of SYT1C2A (left) and E-Syt2C2A (right), Ca²⁺ ions are displayed as white spheres. "N" and "C" stand for N terminus and C terminus, respectively. **(C)** Amino acid sequence alignment of the C2 domains of plant SYTs and human E-Syts. Proteins of known structure are marked with an asterisk and the corresponding Protein Data Bank codes are indicated. Residues involved in Ca²⁺ binding are indicated with a red triangle and highlighted in a red box; residues likely to make up the polybasic sites are indicated with a blue triangle and highlighted in a blue box; secondary structural elements of SYT1C2A are indicated as horizonal arrows and "T" stands for residues involved in a β-turn. SYT1C2A point mutations are indicated with vertical arrows.

resting Ca²⁺ concentrations. We were unable to characterize the second Ca²⁺ site using ITC because no significant heat was measured upon titration of SYT1C2A at higher Ca²⁺ concentrations (Fig 2B). Hence, to characterize the second site, we monitored Ca²⁺-binding using a label-free thermal shift assay (nano-Differential Scanning Fluorimetry, nanoDSF) using Tycho NT. 6 (NanoTemper Technologies). The intrinsic fluorescence of C2A at increased concentrations of free Ca²⁺ were recorded at 330 and 350 nm while heating the sample from 35 to 95°C. Using this technique, the variation of the ratio of fluorescence (350/330 nm) versus the temperature reports on the stability of SYTC2A domain or on the formation of a complex with a ligand if a binding event occurs. Accordingly, our data show that there is an increase in the inflexion temperature ($T_i$) between the folded and unfolded states with increasing Ca²⁺ concentration (Fig 2C, upper panel).

This effect is measurable when moving from 30 to 300 $\mu$M, thus demonstrating a Ca²⁺-binding event is initiated in this concentration range. In addition, the variation of the initial fluorescence emission ratio represents the blue-shift in wavelength that is produced as a result of Ca²⁺ binding to the protein (Fig 2C, upper panel). Thus, we can estimate the $K_d$ because the variation of the initial ratio with increasing ligand concentrations tends to follow typical saturation curves of Michaelis–Menten kinetics (Magnusson et al, 2019). Using this approach, we calculated a $K_d$ of 277 ± 24 $\mu$M (Fig 2C, lower panel). The $K_d$ value for the second Ca²⁺ site is quite above those expected to be physiologically significant, as Ca²⁺

stimulation in plants should be in the $\mu$M range (McAinsh et al, 1990). However, the absence of phospholipids often dismisses the Ca²⁺-binding affinity of C2 domains (Fernandez et al, 2001). To rule out a nonspecific effect of Ca²⁺ on intrinsic fluorescence that accounts for the low affinity $K_d$, we prepared a mutated SYT1C2A in which two of the canonical aspartate residues involved in Ca²⁺ binding were replaced by alanines (D276A, D282A; SYT1C2A-DADA). Our data showed that the addition of Ca²⁺ to SYT1C2A-DADA does not produce a shift on intrinsic fluorescence; indicating that the observed Ca²⁺-binding activity is specifically mediated by the Ca²⁺-dependent lipid-binding site (Fig 2C, lower panel). In addition, we prepared two additional point-mutant proteins, SYT1C2A D282A and SYT1C2A E340A, to investigate the Ca²⁺-binding properties at site II and I, respectively. As expected, the elimination of one carboxylate at the structural site II produces a drastic decrease in the Ca²⁺ binding affinity ($K_d$ = 1.8 ± 0.5 mM), which is coupled with a decrease in thermal stability of the protein ($T_i$ = 55°C) and a red-shift in fluorescence emission with respect to the wild type protein that resembles those effects observed for the SYT1C2A-DADA mutant (Fig S2A). Differentially, SYT1C2A E340A doubled the $K_d$ (534 ± 60 $\mu$M), while reducing slightly its thermal stability (Fig S2A). As expected, the addition of Ca²⁺ to SYT1C2B induced no change in the intrinsic fluorescence of the protein (Fig S2B).

We analyzed the structure of SYT1C2A to identify the role of Ca²⁺ ions in lipid binding. Our data show that Ca II interacts with seven protein atoms, whereas Ca I interacts with five (Fig 2A). This

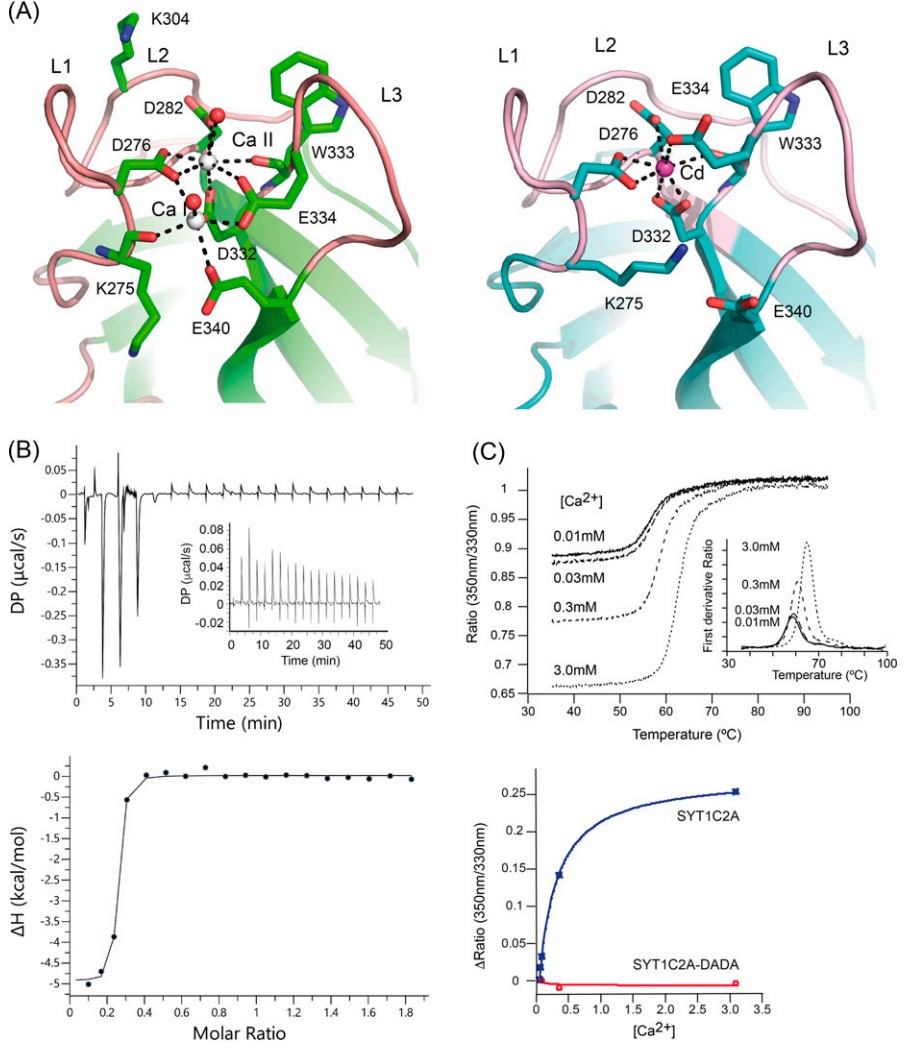

**Figure 2. The structure and properties of the Ca2+–dependent lipid-binding site of SYT1C2A.**
**(A)** A section of the structures of SYT1C2A in complex with $Ca^{2+}$ (left) and $Cd^{2+}$ (Right). **(B)** Calorimetric titration of SYT1C2A with $Ca^{2+}$. (Upper panel) A representative thermogram obtained by the addition of 20 mM $CaCl_2$ to a solution of 151 $\mu$M SYT1C2A at 25°C. The inset corresponds to the thermogram obtained by the addition of $CaCl_2$ to a solution of SYT1C2A supplemented with 10 $\mu$M $CaCl_2$; no detectable heat is observed. (Lower panel) Dependence of the heat released per mol of $Ca^{2+}$ injected as a function of the $Ca^{2+}$:SYT1C2A molar ratio. The solid line corresponds to the best fit of the experimental data based on a one-set-of-sites model. "DP" stands for differential power. **(C)** (Upper panel) Thermal denaturation of SYT1C2A at different $Ca^{2+}$ concentrations. The inset represents the $T_i$ determined as the maximum of the second derivative of the ratio between the fluorescence emission at 350 and 330 nm. (Lower panel) Determination of the $K_d$ of SYT1C2A or SYT1C2A-DADA and $Ca^{2+}$ using the initial fluorescence change as a function of free $Ca^{2+}$ concentration. Three independent replicates were performed.

indicates that Ca II is more tightly bound to the C2 core structure and supports that site II plays a structural function, whereas site I represents the lower affinity $Ca^{2+}$-binding site. This suggests that site I will be occupied depending on the physiological $Ca^{2+}$ concentration to activate the protein and trigger $Ca^{2+}$-dependent SYT1C2A lipid binding.

Unexpectedly, we were able to solve the structure of SYT1C2A with one cadmium ion ($Cd^{2+}$) at site II using different crystallization conditions (Fig 2A and Table S1). This structure mimics that with one $Ca^{2+}$ at the high-affinity binding site. Indeed, the analysis of oxygen, $Ca^{2+}$, and $Cd^{2+}$ complexes found in the Cambridge Structural Database (Groom et al, 2016) of chemical structures determined at high resolution reveals that the coordination numbers and geometries of $Ca^{2+}$ and $Cd^{2+}$ with oxygens are very similar (Fig S3). Interestingly, the comparative analysis of the structures of SYT1C2A in complex with $Ca^{2+}$ and $Cd^{2+}$ shows that Glu 334 at L3 changes its conformation to replace a water molecule at site II, whereas the positively charged side chain of Lys 275 replaces the Ca I (Fig 2A). In this conformation, SYT1C2A might be unable to bind membranes through this site as Ca II would be occluded to interact with the

phosphate moiety of phospholipids. Thus, these data might provide the structural basis for $Ca^{2+}$-mediated activation of SYT1C2A to bind membranes.

## SYT1C2A binds inositol phosphates (IPs) in a $Ca^{2+}$-independent manner

It has been shown that a fragment including C2A and C2B domains (C2AB) of SYT1 is able to bind PIPs (Pérez-Sancho et al, 2015); however, the PIP-binding site has not been described yet. The molecular nature of this interaction and whether or not it occurs through the C2A domain or the C2B remains unknown. In human E-Syts, the interaction between $PI(4,5)P_2$ and some of the C2 domains occurs through a surface-exposed basic patch known as the polybasic site (Giordano et al, 2013; Idevall-Hagren et al, 2015). This binding site is independent of $Ca^{2+}$, and it is built by a number of aromatic and positively charged amino acid side chains placed in the concave side of the $\beta$-sandwich structure of the domain (Guerrero-Valero et al, 2009; Guillén et al, 2013; Diaz et al, 2016; Pemberton & Balla, 2018).

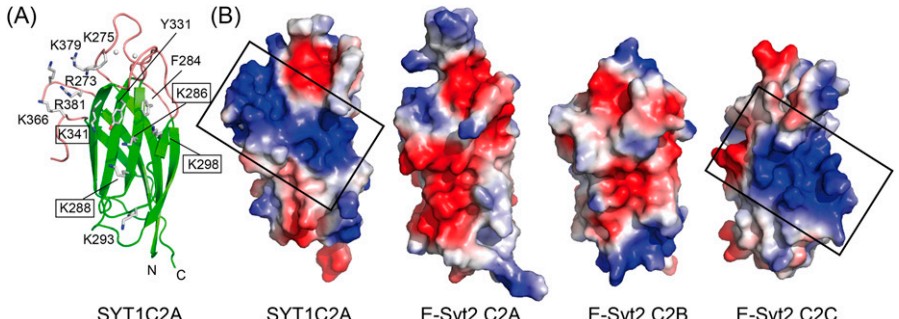

**Figure 3. The polybasic-binding site of SYT1C2A and equivalent sites of C2 domains of E-Syt1.**
**(A)** Ribbon representation of the structure of SYT1C2A showing as sticks the side chains of those residues making up the polybasic binding site. Those residues mutated to prepare the SYT1C2A-PolyB mutant protein are highlighted. **(B)** Surface representation displaying the molecular electrostatic potentials of the SYT1C2A domain and C2 domains of E-Syt2. Red and blue stand for negative and positive potentials, respectively. All the representations have been scaled to the same level. The domains are orientated as the ribbon representation. The polybasic patch of SYT1C2A and E-Syt2C2C are squared. The Protein Data Bank codes for E-Syt2 C2 domains are 4NPJ for C2A and C2B and 2DMG for C2C.

We analyzed the structure of SYT1C2A to explore whether the C2A domain displays a polybasic site to bind PIPs and found a wide-open basic cavity at β3, β2, β5, and β6 sheets that could accommodate large polar heads such as that of PI(4,5)P$_2$. Moreover, the basic characteristic properties of β2 and β3 at this cavity are reinforced by the positively charged amino acids of loop L4 between β7 and β8 (Figs 3 and 1C).

To investigate whether the reported PIP binding to C2AB occurs at SYT1C2A or SYT1C2B and whether it is specifically mediated through the lipid polar head of inositol 1,4,5-trisphosphate (IP$_3$) or it is just an effect of the unspecific binding of SYT1 to a negatively charged bilayer, we investigated the binding of IP$_3$ to SYT1C2A and SYT1C2B using nanoDSF (Table 1). To discard any Ca$^{2+}$-mediated binding, the experiments were performed in the presence of 20 μM EGTA. Our data indicate that IP$_3$ binds to SYT1C2A and not to SYT1C2B because the addition of 58 μM IP$_3$ to SYT1C2A increases the $T_i$ by more than 5° and no increment in $T_i$ is observed when the ligand is added to SYT1C2B (Table 1).

To corroborate that the binding for IP$_3$ occurs at the polybasic site of SYT1C2A, we generated a K286A/K288A/K298A/K341A quadruple mutant protein (SYT1C2A-PolyB) (Figs 1C and 3A). These residues were chosen because they contribute to the basic character of the polybasic site and equivalent mutations in the C2 domain of PKC-α abolish lipid binding to this site (Ferrer-Orta et al, 2017). The negligible effect on the $T_i$ of SYT1C2A-PolyB upon IP$_3$ addition shows that the ligand binds to the polybasic site (Table 1).

Next, we compared the effect of IP$_3$ on SYT1C2A stability with the effect of inositol, inositol 1,4-biphosphate, inositol 1,3,4-trisphosphate, inositol 1,4,5,6-tetraphosphate, and 3,4,5,6-tetraphosphate to evaluate the specificity of SYT1C2A for IPs. Our data show that the pocket is not specific for any of the phosphorylated inositol molecules tested because all of them bind equally to SYT1C2A. In stark contrast, inositol did not produce an increase of the $T_i$, which clearly indicates the requirement of a phosphate moiety to bind SYT1C2A. None of the phosphorylated inositol molecules tested was able to bind SYT1C2A-PolyB (Table 1).

Finally, we analyzed the charge distribution at the molecular surface of those C2 domains from E-Syts with known three-dimensional structures (Xu et al, 2014) (Fig 3B). The analysis clearly shows that SYT1C2A and E-Syt2C2C, which have been reported to bind PI(4,5)P$_2$, display similar surface properties at the polybasic site. Conversely, E-Syt2C2A and E-Syt2C2B, which are unable to bind them, do not display the site (Fig 3B) (Giordano et al, 2013; Idevall-Hagren et al, 2015; Bian et al, 2018). Altogether, our data indicate that the polybasic site of SYT1C2A determines the PI(4,5)P$_2$-binding properties of SYT1.

The comparison of the protein sequence of SYT1C2A with those from other C2 domains from plant SYTs whose structures are still unknown (Fig 1C) shows that the overall architecture of the polybasic site of the SYT1C2A domain should be preserved in other C2A of plant SYTs, as most of the basic residues of β2 and β3 and the insertion at L4 are conserved. Hence, in contrast to the E-Syts, the plant C2A domains are likely to play an active role, providing specificity to the interaction of SYTs with PIPs at the PM. Conversely, as expected from our biochemical data, SYT1C2B and its protein orthologs do not harbor the basic residues that confer PI(4,5)P$_2$-binding activity.

### SYT1C2A binds phosphatidylserine (PS) and PI(4,5)P$_2$ to cooperatively bind model membranes

Our structural and biochemical data support a membrane binding model in which SYT1C2A binds to PS through the Ca$^{2+}$-dependent lipid-binding site and to PI(4,5)P$_2$ through the polybasic site. In contrast, our results indicate that the reported membrane-binding property of SYT1C2B (Schapire et al, 2008) is constitutive as this protein fragment binds neither Ca$^{+2}$ nor IPs. To corroborate these data, we analyzed the membrane-binding properties of SYT1C2A and SYT1C2B fragments using lipid sedimentation assays. We monitored the ability of these fragments to bind liposomes

**Table 1. Inflexion temperature of SYT1C2A and SYT1C2B in the presence 20 μM EGTA and different lipid polar heads.**

| Lipid polar heads | SYT1C2A | C2APolyB | SYT1C2B |
|---|---|---|---|
| Control | 57.4 ± 0.5 | 61.4 ± 0.1 | 51 ± 1 |
| IP2 | 59.4 ± 0.2 | 61.5 ± 0.1 | — |
| IP3 | 62.60 ± 0.05 | 61.4 ± 0.1 | 51 ± 1 |
| IP(1,3,4) | 64.2 ± 0.1 | 61.4 ± 0.3 | — |
| IP(1,4,5,6) | 62.6 ± 0.2 | 61.5 ± 0.3 | — |
| IP(3,4,5,6) | 65.2 ± 0.4 | 61.4 ± 0.3 | — |
| Ins | 56.8 ± 0.2 | — | — |

IP2, inositol 1,4-biphosphate; IP3, inositol 1,4,5-trisphosphate; IP(1,3,4), inositol 1,3,4-trisphosphate; IP(1,4,5,6), inositol 1,4,5,6-tetraphosphate; IP(3,4,5,6), inositol 3,4,5,6-tetraphosphate; Ins, inositol. The unbound protein is taken as a control.

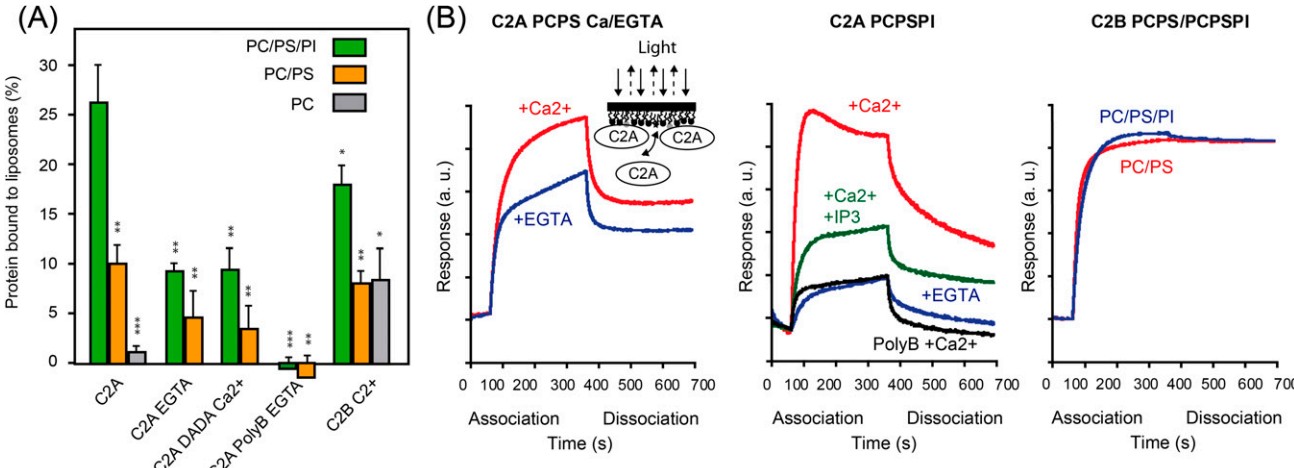

**Figure 4. Lipid-binding properties of SYT1C2A.**
**(A)** Comparative analyses of phospholipid binding of SYT1C2A and mutants. Protein quantifications of the soluble fraction after lipid pelleting were performed by measuring the intrinsic Trp fluorescence under denatured conditions. Lipid binding activity is expressed as the percentage of the bound protein to lipids. Error bars indicate the standard error calculated from three independent measurements. **(B)** Representative sensograms representing the binding of SYT1C2A, SYT1C2A-PolyB, and SYT1C2B to a PC/PS or to a PC/PS/PI monolayers in the presence of $Ca^{2+}$, EGTA, or IP3. "a.u." stands for arbitrary units. The inset represents a scheme of a bio-layer interferometry biosensor in a SYT1C2A bound state. As molecules bind the surface, the layer thickens at the end of the tip and the path length of the reflected light changes. $t$ test ($P < 0.5*$; $P < 0.1**$; $P < 0.05***$).

composed of 75% phosphatidylcholine (PC) and 25% PS (PC/PS) 70% PC, 25% PS, and 5% PI(4,5)$P_2$ (PI) (PC/PS/PI) or 100% PC.

Our data showed that SYT1C2A binds PC/PS and PC/PS/PI membranes in a $Ca^{2+}$-dependent manner, as the amount of protein bound to liposomes decreases in the presence of EGTA and the SYTC2A-DADA mutant reduces the binding to the same extend. The interaction is further reduced when the SYT1C2A-PolyB mutant is assayed in presence of EGTA. These data indicate that the $Ca^{2+}$ and polybasic-binding sites cooperate in the binding to model membranes. Remarkably, we observed a reduction in the binding properties of SYT1C2A when comparing PC/PS and PC/PS/PI membranes, suggesting an effective role of PI(4,5)$P_2$ in protein membrane interaction. In addition, our results corroborate previous findings (Schapire et al, 2009) showing that SYT1C2A binds negatively charged lipids and does not bind to PC (Fig 4A).

Similar results were obtained using bio-layer interferometry (BLI) to monitor the lipid-binding properties of the C2 domains of SYT1 (Sultana & Lee, 2015). In this assay, we immobilized lipid monolayers composed of a mixture of 75% and 25% PS (PC/PS) or 65% PC 25% PS 10% PI(4,5)$P_2$ (PC/PS/PI) on a biosensor tip. As SYT1C2A or SYT1C2B binds to lipids, incident light directed through the biosensor shifts and creates a quantifiable interference pattern (Fig 4B). Monitoring protein interaction with the tip in real time provides kinetic data on molecular interactions that can be used to determine relative affinities. Our data confirm that SYT1C2A binds PC/PS membranes in a $Ca^{2+}$-dependent manner as we observed a reduction in the amount of protein bound to the sensor tip in the presence of EGTA. The same effect by EGTA was observed when using PC/PS/PI monolayers, but this reduction was intensified when the experiments were performed using the SYT1C2A-PolyB mutant protein or in the presence of IP$_3$. These results support the involvement of the polybasic site in protein SYT1C2A interaction with the PC/PS/PI monolayer as IP$_3$ would compete with PI(4,5)$P_2$ for the polybasic-binding site.

Our data also confirm that SYT1C2B binds model membranes, remarkably, this includes the neutral PC liposomes (Fig 4A and B). The BLI experiments showed that the protein remains anchored to the biosensor tip during the dissociation step (Fig 4B, right panel), thus supporting that SYT1C2B displays a constitutive membrane-binding mode.

### The structure of SYT1C2A in complex with PS and IP$_3$

We modeled the complexes of SYT1C2A with PS and with PI(4,5)$P_2$ to unravel the structural basis of the domains' lipid specificity. To build these structures, we took advantage of our biochemical data and the structural knowledge about this interaction with other C2 domains. Among them, the complex between dicaproylphosphatidylserine (DCPS) and the C2 domain PKC at the $Ca^{2+}$-dependent lipid-binding site (Verdaguer et al, 1999) and the complexes between inositol phospholipids and PKC (Guerrero-Valero et al, 2009) or Rabphilin (Guillén et al, 2013; Ferrer-Orta et al, 2017) at the polybasic site. Hence, we superimposed those structures to that of SYT1C2A to identify the lipid-binding sites and modeled DCPS and IP$_3$ in complex with SYT1C2A using the GOLD software (Jones et al, 1997). We included as an additional binding site the basic patch located in the vicinity of L4 (Fig 3) as it displays the size and electrostatic potential that are necessary to accommodate an IP$_3$ molecule. The protein was treated as rigid and full flexibility was allowed for the ligands. We selected meaningful interactions through visual inspection of the top-scoring ligand solutions using molecular graphics.

The lipid specificity of the $Ca^{2+}$-dependent lipid-binding site is restricted by the conformation of loops L1, L2, and L3 (Verdaguer et al, 1999). Thus, the narrow cavity defined in SYTC2A at this site may explain the $Ca^{2+}$-dependent selectivity for a lipid with a small polar head such as PS (Schapire et al, 2009). Accordingly, the DCPS molecule docks into the $Ca^{2+}$-dependent lipid-binding site making

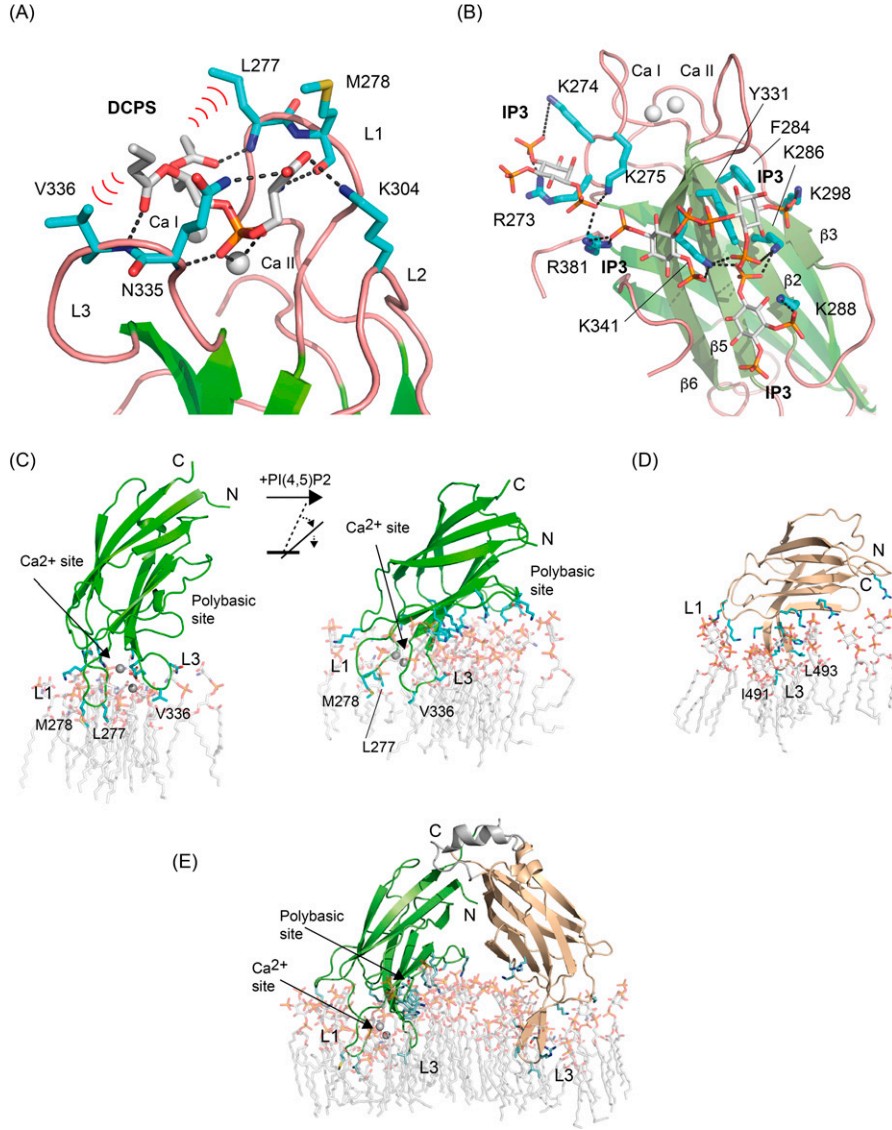

**Figure 5. SYT1C2A recognizes PS and phosphorylated inositol lipids.**
**(A, B)** Computational model of the $Ca^{2+}$-dependent and polybasic lipid-binding site in complex with DCPS and IP3, respectively. The predicted hydrogen bond and polar interactions between ligands and protein are shown as dashed lines. Relevant hydrophobic interactions are highlighted with red waves. **(C)** Comparison of the association of SYT1C2A to the PS-M (left) and to the PSPIP-M (right). **(D, E)** The association of SYT1C2B and SYT1C2AB to the PSPIP-M, respectively. The residues and lipids involved in protein membrane interaction are shown as sticks.

several favorable interactions (Fig 5A). The model shows that the phosphate group coordinates Ca II, whereas the hydrophobic moiety of DCPS interacts with Leu 277 and Val 336 at the solvent-exposed tips of L1 and L3. Conversely, the lipid polar head points towards L2 where the presence of the positively charged Lys 304 at the entrance of the cavity may hinder the interaction of SYT1C2A with positively charged lipid polar heads.

As expected, the IP$_3$ molecule docks into the polybasic site of SYT1C2A and establishes favorable electrostatic interactions with the $PO_4^{2-}$ moieties. In addition, aromatic Phe 284 and Tyr 331 interact with the hydrophobic face of the inositol ring. However, the molecule does not bind at a unique defined site; rather, it is predicted to bind at different places that have in common the position of the phosphate moieties of IP$_3$. These groups are inserted into a small positively charged pocket defined by a network of Lys and Arg residues placed regularly on the concave side of SYT1C2A (Fig 5B). In accordance with our biochemical data, this network would not restrict the interaction to any specific phosphorylated inositol and presages an efficient interaction with them.

### The topology of SYT1C2A membrane association depends on $Ca^{2+}$ and PI(4,5)P$_2$

The topology and orientation of the C2 domains with respect to the membrane and the membrane penetration are determined by the restrictions imposed by the binding to the $Ca^{2+}$-dependent or to the polybasic lipid binding sites. Different experimental and computational approaches to this issue have led to the conclusion that the C2 domains may associate with membranes in at least two possible manners (Cho & Stahelin, 2006; Ausili et al, 2011; Honigmann et al, 2013; Prasad & Zhou, 2020): in the first conformation, the domain is oriented perpendicularly to the plane of the membrane, with the $Ca^{2+}$-binding site interacting with the polar heads of the membrane lipids; and in the second, the C2 domain is

oriented parallel to the membrane, with both the Ca$^{2+}$-binding site and the polybasic site interacting with the membrane.

To elucidate how SYT1C2A associates to the membrane and to provide mechanistic understanding of this interaction, we performed unrestrained molecular dynamics (MD) simulations of a Ca$^{2+}$-bound SYT1C2A molecule attached to a model membrane containing either solely PS and sitosterol (PS-M) in both monolayers or a randomly distributed mixture of PS, sitosterol, and PI(4,5)P$_2$ in the lower leaflet (PSPIP-M, see the Materials and Methods section). Both simulations revealed a stable association of SYT1C2A (see the Materials and Methods section) with the inner leaflet that was driven by distinct interactions of the two bound Ca$^{2+}$ ions with the PS head groups (Figs 5C and S4A and Video 1). Remarkably, the incorporation of PI(4,5)P$_2$ to the inner leaflet translated into an induced tilt of the domain that led SYT1C2A to adopt a more parallel arrangement with respect to the plane of the membrane (Figs 5C and S4B and Video 1). This tilt was triggered by the progressively tighter electrostatic interactions between the polybasic-binding site of SYT1C2A and the PI(4,5)P$_2$ head groups (Fig S4C). From both simulation results, we conclude that both conformations are feasible and dependent on the presence of PI(4,5)P$_2$ in the inner leaflet of the membrane.

The analysis of the perpendicular conformation of SYT1C2A bound to the PS-M reveals that the Ca$^{2+}$-dependent association encompasses the insertion of the hydrophobic tip of loops L1 (Leu 277 and Met 278) and L3 (Val 336) with the formation of direct hydrogen bonds between most of the polar residues in these loops with the polar heads of ~14 PS molecules. A similar pattern of interactions in the vicinity of the Ca$^{2+}$-dependent binding site is observed for the parallel orientation of SYT1C2A bound to PSPIP-M. However, loops L1 and L3 penetrate deeper into the membrane and form an increased number of direct hydrogen bonds between SYT1C2A residues and the lipids, of which 1 is sitosterol, ~8 are PS and ~10 are PI(4,5)P$_2$. The observed number of hydrogen bonds between SYT1C2A and the membranes is conserved along the MD simulation (Fig S4A and B). Of particular interest are the interactions observed between Val 336, Leu 277, and Met 278 and PS (Fig S4D) and the hydrophobic face of the phosphoinositol heads of lipids at the polybasic site, as they interact with Phe 284 and Tyr 331 (Fig S4E) in a fashion similar to that observed for the interaction between phosphoinositides and the C2 domain of PKC (Guerrero-Valero et al, 2009) and the modeled complex between SYT1C2A and IP$_3$ (Fig 5B). These observations provide molecular insight into the enhanced affinity and specificity of SYT1C2A for PI(4,5)P$_2$-containing membranes (Fig 4).

Next, we performed MD simulations on the SYT1C2B fragment attached to a PSPIP-M model membrane to investigate the basis of its membrane binding properties. The simulation revealed a stable association with the membrane that is governed by the insertion of the hydrophobic tip of loops L1 and L3 among the hydrocarbon tails of the phospholipids (Fig 5D). The SYT1C2B domain displays up to nine direct hydrogen bonds with PI(4,5)P$_2$ molecules. However, these interactions with the lipid polar heads do not involve the inositol ring; rather they are charge-based interactions, solely mediated through the PO$_4^{2-}$ moieties.

Finally, we investigated if the pattern of interactions of each individual C2 domain with the PSPIP-M model membrane is

conserved when the MD simulations are performed using the V-shaped conformation of C2AB observed in the crystal structure of E-Syt2 (Schauder et al, 2014; Xu et al, 2014). Our calculations showed that the hydrophobic tips of the Ca$^{2+}$-dependent lipid-binding site of SYT1C2A and loop L3 of SYT1C2B are inserted into the membrane (Fig S4F). Remarkably, the latter is structured into a well-defined β–hairpin whereas the residues making up the polybasic site of SYT2A interact with the polar heads of several PI(4,5)P$_2$ molecules (Figs 5E and S4G). This putative binding mode suggests the feasibility of a stable and lasting interaction of the C2AB tandem with the membrane that depends on Ca$^{2+}$ and PI(4,5)P$_2$ molecular signals.

### SYT1 displays three flexible hinge points

The crystal structure of a E-Syt2 fragment comprising the SMP and the two adjacent C2 domains (C2A and C2B) revealed a modular structure in which the SMP domain dimerizes to form an elongated cylinder of around 90 Å and two rigid tandems formed by independently arranged C2A and C2B domains that are linked to the SMP through a flexible linker (Schauder et al, 2014). The sequence analysis of the SMP and both C2A and C2B domains of SYT1 reveals that they are structurally similar to those of E-Syt2. However, the relative position between the domains cannot be accurately predicted as the aminoacidic sequence linking any two of them is not conserved (Phyre2 server [Kelley et al, 2015]). This arrangement and the flexibility of the amino acid stretch connecting the SMP to the C2AB domains is likely to be related to SYT1 function as it restricts the relative orientation of these domains with respect to the ER and PMs. Consequently, the knowledge derived from the study of the molecular organization of SYT1 will provide information on the molecular function of the protein.

To investigate whether the domain organization observed for E-Syt2 is conserved in SYT1, we analyzed two different overlapping fragments of the SYT1 structure using SAXS: the SMPC2A, from residue 34 to residue 397, and the C2AB, from residue 253 to the C terminus (Fig 1A). This powerful technique yields structural information of both ordered and disordered biological macromolecules at low resolution, including their size and shape in solution as well as their flexibility (Hura et al, 2009).

Analysis of the one-dimensional SAXS experimental curves was initially performed to judge the quality of the data and to obtain basic structural information of the proteins under study (Fig S5). The data indicated that the SMPC2A fragment displays a molecular mass of 73 kD, a radius of gyration ($R_g$) of 44.9 Å and a maximum size (D$_{max}$) of 176 Å. These values are in accordance with a dimeric structure of SYT1. It is worth to note that the size of the SMP tunnel of the E-Syt2 (Schauder et al, 2014) is two times smaller than the Dmax of SMPC2A (see the Materials and Methods section, Table S2).

Protein flexibility in solution can be experimentally qualitatively estimated by the Kratky plot and quantitatively by means of the Porod exponent analysis. This last parameter ranges from two to four for flexible and compact proteins, respectively (Feign & Svergun, 1987; Rambo & Tainer, 2011). The estimated Porod exponent of 2.8 for SMPC2A denotes that the molecule is compact but displays some degree of flexibility. This suggests that the relative position of the two C2A domains with respect to the dimeric SMP domain is variable. To investigate this issue in detail we used a SAXS-based pseudo atomic

modeling approach using the MultiFoXS server (Schneidman-Duhovny et al, 2016). This technique can be used for the structural characterization of flexible proteins in solution if a high-resolution structure or a comparative model of the studied protein is available. The procedure consists in a low-resolution rigid-body fitting to the experimental data that includes flexibility between the folded domains that make up the structure. This is performed by sampling random conformations along flexible residues and considering that an ensemble of multiple conformations contributes to a single observed SAXS profile. Thus, to build a model of the protein in solution we used the high-resolution atomic structure of SYT1C2A and a model based on the atomic structure of the SMP domain from E-Syt2 (Kelley et al, 2015). In addition, we model-built residues N34 to K67 and the linker between the SMP and the C2A domains that were not present in the initial model (see the Materials and Methods section). The server sampled more than 10,000 conformations, calculated their SAXS profiles, and scored multi-state models according to their fitting to the experimental profile. The output reveals that the fitting of a two-state model leads to a significant improvement relative to the model based on the X-ray structure (Fig S6). The analysis of this manifold conformational model confirms that the linker between SMP and C2A is indeed a flexible hinge point (Fig 6A and B). In addition, the N-terminal extension of SMPC2A that links the hydrophobic segment inserted into the ER membrane with the SMP domain (residues from N34 to K67) (Fig 6A) remains unstructured and is likely to represent an additional hinge point in the molecule. Finally, the low-resolution structure shows a symmetric and extended arrangement of the C2A domains with respect to the SMP domain that differs from the compact conformation observed in the crystal structure of human E-Syt2.

The SAXS data corresponding to the C2AB fragment indicate a molecular mass of 28 kD, an $R_g$ of 28.2 Å, and a maximum size of 121 Å. These parameters are in accordance with a monomeric and elongated structure of the C2AB domain (Table S2). The Porod exponent for this fragment is 2.0, indicating that the molecule is highly flexible. However, the shaped features displayed by the pair distribution function at long distances (Fig S5) indicate that the fragment may adopt several distinct conformations. To analyze the SAXS data of this construct, we used the same approach used for the SMPC2A fragment, using as a template the high-resolution

atomic structure of the human C2A–C2B fragment (Xu et al, 2014) and defining a hinge point at the linker between the C2 domains. The resulting model includes two independent and largely different conformations that together fit to the experimental data (Figs 6C and S6). Interestingly, whereas one of those conformers displays a V-shaped compact structure that resembles that observed in the crystal structure of E-Syt2 (Schauder et al, 2014; Xu et al, 2014), the other shows an extended conformation in which there is no interaction between the C2A and C2B domains (Fig 6C). Interestingly, the V-shaped structure in solution resembles the proposed binding mode for SYT1C2AB to the membrane. The analysis of the superposition of these low-resolution models confirms the presence of an additional hinge point at the linker between the C2A and the C2B that contrasts with the crystallographic data of E-Syt2, in which the C2A and C2B domains are arranged as a rigid tandem that is not altered by the presence of the SMP domain or the $Ca^{2+}$ ions (Schauder et al, 2014; Xu et al, 2014). Interestingly, our analysis of the data obtained from the SAXS experiments performed in the presence of either 1 mM $Ca^{2+}$ or 1 mM EGTA did not reveal any significant differences (Figs S5 and S6 and Table S2).

# Discussion

### SYT1C2A domain facilitates SYT1 response to $Ca^{2+}$ and PI(4,5)P2

The molecular mechanisms controlling the membrane tethering and lipid transport activities of E-Syts is tightly regulated by the accumulation of PI(4,5)P$_2$ at the membrane and by the elevation of cytosolic $Ca^{2+}$ concentration (Giordano et al, 2013; Idevall-Hagren et al, 2015; Saheki et al, 2016; Yu et al, 2016; Bian et al, 2018; Lee et al, 2019). This regulation, in turn, is encoded in a complex system of domain–domain and domain–membrane interactions in which the different properties of the multiple C2 domains comprised in the E-Syts structure play a central role. This diversity is encompassed in the two C2 domains of plant SYTs. In vivo studies showed that SYT1 is anchored to the ER through an N-terminal hydrophobic region and binds to the PM through the two C2 domains at the C-terminal end (Pérez-Sancho et al, 2015). It has also been shown that, whereas

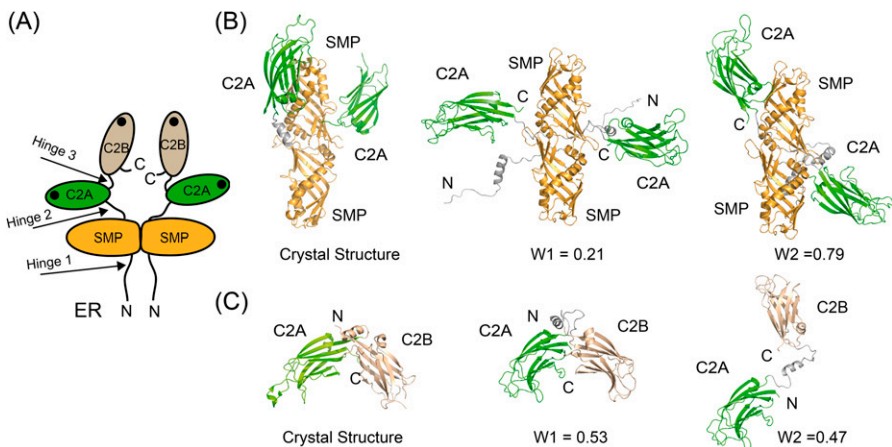

**Figure 6. Low resolution structure of SYT1.**
**(A)** Schematic representation of the structure and topology of SYT1. The black dots represent the side where membrane binding occurs. **(B)** Comparison of the ribbon representation of the SMPC2A fragment of E-Syt2 crystal structure with two representative conformations of SYT1 SMPC2A fragment in solution. **(C)** Comparison of the ribbon representation of the C2AB fragment of E-Syt2 crystal structure with two representative conformations of SYT1C2AB fragment in solution. The relative weight of each conformer as estimated by the MultiFox server is indicated (W1 and W2).

the C2A domain displays Ca²⁺-dependent activity to bind model membranes, the C2B domains display constitutive membrane-binding properties (Schapire et al, 2008). These data suggest that C2B is likely to play a passive role for the tethering activity of SYT1, whereas C2A may additionally regulate the lipid transport activity. To investigate the membrane-binding properties of SYT1C2A, we have determined its 3D structure and analyzed its Ca²⁺- and lipid-binding properties.

Our structural data show that SYT1C2A displays a Ca²⁺-binding site with high affinity which, even at resting Ca²⁺ concentrations, is occupied and inaccessible to solvent, and another, which functions as a relay for sensing physiological Ca²⁺ variations and triggers the Ca²⁺-dependent binding to lipids (Fig 2). Similar Ca²⁺-binding properties have been reported for the C2A domain of human E-Syt2 (Xu et al, 2014), as well as for other C2 domains such as human synaptotagmin (Fernández-Chacón et al, 2002), dysferlin (Fuson et al, 2014), or for the *Arabidopsis* CAR proteins (Diaz et al, 2016). Thus, it is likely that this mechanism is shared by all of these C2 domains.

In plants, CS expansion mediated by SYTs in response to stress requires two events, namely, the activation of Ca²⁺ signaling and the accumulation of PIP at the PM. However, the dynamics of such a process is consistent with the slow accumulation of PIP rather than with the fast increase of Ca²⁺ concentration (Lee et al, 2019, 2020). Our structural and biochemical data show that SYT1C2A provides a framework to reconcile these observations. We have shown that the domain displays a dual activity to bind PS or PIPs in a Ca²⁺-dependent or independent manner, respectively. These interactions cooperate to enhance the SYT1C2A affinity for membranes (Fig 4) as they occur at different sites: a polybasic site to accommodate PIP interaction and a Ca²⁺-dependent lipid binding site to bind PS (Figs 3 and 5). Together, these data imply that the Ca²⁺-dependent interaction with PM of SYT1C2A occurs first and thereafter PIP triggers CS expansion. In contrast, our data further support the available evidence related to the non-regulated tethering function of SYT1C2B (Schapire et al, 2008), as we have confirmed that SYT1C2B binds model membranes (Fig 4) and have found that the protein does not bind Ca²⁺ nor IP₃ (Fig S2 and Table 1).

Interestingly, we have shown that SYT1C2A binds IP₃ as well as other phosphorylated inositol derivatives (Table 1). These interactions might compete with those involving PIPs in living cells. For instance, the protein kinase PDK1 interacts with Ins(1,3,4,5,6)P₅ and Ins(1,2,3,4,5,6)P₆ to prevent PIP-mediated interaction to membranes and to retain the protein in the cytosol (Komander et al, 2004). An

increase in IP₃ levels may result from the hydrolysis of the PIPs to produce DAG because of the activation of the phospholipase C activation in response to stress (Saheki et al, 2016; Bian et al, 2018). Moreover, it has been proposed that the role of SYT1 and SYT3 in plants—and E-Syts in mammals—is to transfer DAG from the PM to the ER to maintain membrane stability (Saheki et al, 2016; Ruiz-Lopez et al, 2020 *Preprint*). Thus, our data provide insight for the regulation of SYTs in response to IP₃ and DAG production at membrane CS.

### The polypeptide chain of SYT1 provides the molecular flexibility required for tether and lipid transfer functions

In *Arabidopsis*, abiotic stress produces PM instability by the accumulation of DAG. SYT1 and SYT3 act as molecular tethers between the ER and the PM and participate in the restoration of PM homeostasis by transporting DAG to the ER (Lee et al, 2019; Ruiz-Lopez et al, 2020 *Preprint*). Despite this knowledge, the molecular mechanism by which SYTs associates with PM to extract and transfer DAG is unclear. The available structural and biochemical data suggest that the SMP dimer (i) is tethered to the ER through the N-terminal hydrophobic segment and to the PM trough the C2 domains and (ii) shuttles lipids from one membrane to another (Saheki & De Camilli, 2017a, 2017b; Wang et al, 2017; Wong et al, 2018) (Fig 7).

The lipid transfer activity of E-Syts necessarily requires conformational changes that enable the loading of lipids from the PM to the SMP domain and their delivery to the ER or vice versa. Moreover, protein–membrane interactions and the alteration of local lipid composition triggered by stress imply changes in the membrane that the protein should accommodate. The structure of SYT1 at low resolution revealed three flexible hinges within the protein (Fig 6). Our SAXS data show that this feature enables SYT1 to undergo conformational changes that involve increases in the calculated $R_g$ of 34% for the C2AB fragment (2.3–3.5 nm) and 8% for the SMPC2A fragment (3.8–4.1 nm) (Fig 6 and Table S2). This capacity provides a high degree of conformational freedom that facilitates macromolecular motions to adapt to the changing composition of PM in response to stress and to trigger lipid transport.

The lipid-binding properties of the C2 domains of SYT1 suggest that SYT1C2A might provide the driving force for the functional reorganization of the protein at membrane CS. In the resting state, at low Ca²⁺ concentrations, SYT1 displays a broad localization in the ER although it is enriched at CS (Pérez-Sancho et al, 2015). In this

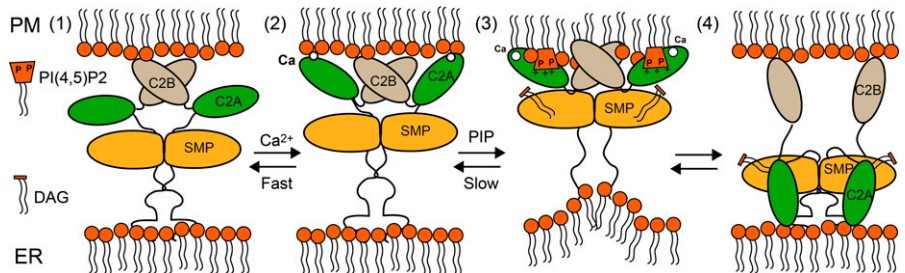

**Figure 7. Schematic representation of the interaction of SYT1 with the ER and plasma membrane (PM).** (Step 1) In the resting state, SYT1 is anchored to the ER membrane through the N-terminal hydrophobic segment and to the PM through the C2B domain. Stress elicits a fast Ca²⁺ signal (Step 2) followed by a slow accumulation of PI(4,5)P₂ at the PM (Step 3). This triggers the binding of C2A to the PM. Molecular flexibility provides de basis for intimate adaptation of the C2 domains to the PM and enables a strong interaction. This rearrangement may generate tense molecular conformations and/or membrane distortions that would favor lipid extraction from the membrane and loading to SYT1. The PI(4,5)P₂ reduction at PM might enable the binding of C2A to the ER membrane. This may represent the end point of the cycle in which DAG is delivered to the ER (Step 4).

situation, SYT1C2B may be tethering ER and PM, whereas SYT1C2A could be detached from the PM (Fig 7, step 1). The Ca$^{2+}$-dependent interaction of the SYT1C2A domain with the PM will promote the reported fast CS expansion by the enhancement of the tethering function of SYT1 (Pérez-Sancho et al, 2015; Lee et al, 2019). The SAXS data showed that the maximum molecular dimension of the SMPC2A fragment does not differ much from the distance between the ER and the PM in yeast's CS (Collado et al, 2019) (17 and 23 ± 5 nm, respectively). Indeed, it has been shown that overexpression of Syt1 induces a decrease of ~7 nm in the intermembrane distance of CS (Fernández-Busnadiego et al, 2015). Hence, it is plausible that SYT1 tethers both membranes if the protein displays an extended conformation with its long axis perpendicular to the membrane, as shown here by MD simulations in the presence of Ca$^{2+}$ (Fig 5). In addition, this molecular expansion might reshape the interface of SYT1C2A with the SMP domain, so as to enable lipid loading (Figs 6 and 7 step 2).

Further conformational changes are expected in SYT1 because the binding of SYTC2A to PIPs is coupled to a change in the relative position of SYT1C2A with respect to the plane of the membrane, that is, from a perpendicular to a parallel orientation because of the geometrical restraints imposed by the occupation of the polybasic lipid-binding site (Fig 5).

SYT1C2A may operate independently of the SMP domain by means of the unstructured hinge sequence between them; however, there is no evidence that the SMP dimers associate with membrane in vivo (Reinisch & De Camilli, 2016) or in vitro (Ge et al, 2021); hence, this transition could be coupled to the approach of the SMP domain to the PM for lipid loading because the lipid-binding sites and the N-terminal linker to the SMP are located on opposite sides of the C2 domain (Fig 7 step 3).

Finally, the variable organization of the C2A domain with respect to the C2B domain contrasts with the rigid architecture of the C2AB tandem of human E-Syt2 and better resembles that observed for the C2A and C2B domains of synaptotagmin (Fig 6). The latter enables the simultaneous and strong binding of C2A and C2B to the membrane and provides functional versatility to bridge vesicles and PM before membrane fusion (Prasad & Zhou, 2020). Similarly, one may consider the possibility that SYT1C2AB could be anchored to the PM for lipid loading or could be bridging ER and PM for lipid release to the ER. Indeed, C2A binding to the ER membrane should also be considered as it contains phospholipids with a net negative charge such as PS (Fouillen et al, 2018). This may represent the end point of a model in which DAG could be delivered to the ER (Fig 7 step 4). Because the cis- and trans membrane interactions of Syt1 in vitro are driven by the balance of anionic lipids between target membranes and by the Ca$^{2+}$ and protein concentrations (Vennekate et al, 2012), the changes in the local composition of PIP and/or Ca$^{2+}$ may also drive the transfer of SYT1C2A from the PM to de ER and vice versa, whereas C2B remains at the PM.

### The structure of SYT1 shares features of proteins that induce membrane instability

The asymmetrical insertion of a protein fragment into one leaflet of the lipid bilayer induces local membrane destabilization and generates curvature (McMahon & Boucrot, 2015; McMahon et al, 2010). As a consequence, the energy barrier required for lipid extraction or insertion from or to the bilayer is reduced (Collado et al,

2019). Indeed, it has been suggested that the insertion of the N-terminal hydrophobic end of Tcbs may induce the formation of peaks of strong curvature at the ER region facing the PM. These structures shorten the distance between the ER and the PM by ~7 nm and facilitate lipid transport (Collado et al, 2019). The N-terminal hydrophobic end of human E-Syts and Tcbs is folded as a helical hairpin inserted into one leaflet of the ER membrane (Giordano et al, 2013; Saheki & De Camilli, 2017b). The comparison of the hydrophobic segments of these proteins and those of plant SYTs shows that they are longer than single transmembrane helices of proteins inserted into the ER membrane (34 and 35 amino acids for SYT1 and SYT3, respectively, and 20 amino acids for a transmembrane helix) (Sharpe et al, 2010) (Fig S7). In addition, the HHpred server (Zimmermann et al, 2018) predicts a remote homology with the structure of a membrane hairpin with two helices that face the N- and C-termini towards the cytosol (Howell et al, 2005). In this structure, a conserved pattern of Gly residues, which is also observed in SYTs, facilitates the packing of the helices making up the hairpin and the formation of the loop connecting them. Thus, it is likely that SYTs are also anchored to the ER membrane by a hairpin that plays an active role in the function of SYT1 by reducing the distance between the ER and the PM and facilitating lipid transfer, as observed in Tcbs.

The Ca$^{2+}$-dependent shallow membrane insertion of the loops making up the Ca$^{2+}$-dependent lipid-binding site at the level of the glycerol backbones of the phospholipids induces membrane instability (McMahon & Boucrot, 2015). Our MD simulations show that membrane binding of SYT1C2A involves the insertion of the L1 and L3 tips to this level; moreover, this insertion is more pronounced and distorts membrane structure when the domain is orientated in a parallel conformation and when the SYT1C2B is also considered (Figs 5C and E and S4G). These data suggest that the regulated binding of the SYT1C2A contributes to membrane destabilization and promotes lipid extraction. Since the aligned protein sequences of those C2 domains that display a Ca$^{2+}$-binding signature conserve the hydrophobic residues at the tips of loops L1 and L3, it is reasonable to assume that they all share a similar insertion mode (Fig 1C).

The differences in size and shape of the lipids making up the inner and outer monolayers are a source of membrane instability. The accumulation of lipids endowed with bulky head groups, such as PI(4,5)P$_2$, favors positive curvature as they act as wedges. Conversely, DAG imposes negative curvature as it has no polar head (Di Paolo & De Camilli, 2006). E-Syts are regulated by the accumulation of PIPs in the PM and their function is to clear away the DAG formed by the action of phospholipase C (Saheki et al, 2016; Ruiz-Lopez et al, 2020 Preprint). This suggests that the clearance of DAG, by itself, may act as a regulator of protein function as this process will contribute to membrane stabilization and hinder lipid extraction.

Taking all of our results together, we propose a working model by which the flexible SYT1 molecule integrates PIP and Ca$^{2+}$ signals elicited by stress through the robust interaction of the SYT1C2A domain with the PM. This process may generate membrane instability and/or tense conformations that could favor lipid transport from the PM to the ER. These findings represent a substantial progress towards elucidating the basis for plant adaptation to stress and may have biotechnological implications. In addition, our data provide insight into a fundamental process common to all

eukaryotic cells. Human E-Syts, yeast tricalbins, and plant SYT1 share both molecular structure and function and, thus, it is likely that they also conserve the molecular basis of their membrane-delimited regulatory mechanisms.

# Materials and Methods

### Protein expression and purification

The pGEX6P1:SYT1C2AB (residues K253-Cterm), pGEX6P1:SYT1C2A (residues K253-E397), pGEX6P1:SYT1C2A-DADA (D276 and D282 into A), pGEX6P1:SYT1C2A-PolyB site mutant (K286, K288, K298, and K341 into A), pGEX6P1:SYT1C2B (residues G374-end), and pGEX6P1:SYT1-SMPC2A (residues N34-E397) were transformed into Rosetta 2 (DE3) pLysS cells (Stratagene) for protein expression using standard protocols. The same protocol of expression was used for the three proteins. A total of 48 ml of a 6-h culture was subcultured into 6 liters of fresh 2TY broth (16 g Bacto tryptone, 10 g yeast extract, 5 g NaCl/liter of solution) plus ampicillin (100 $\mu$g/ml) and chloramphenicol (35 $\mu$g/ml) at 37°C. When the OD at 600 nm was 0.9, the overnight protein expression was induced with 0.3 mM isopropyl-$\beta$-d-thiogalactoside at 16°C. Cells were harvested by centrifugation (30 min, 4,750 rpms in a SX4750 Beckman Coulter swinging bucket rotor) at 4°C and all further handling of sample was done at this temperature. Pellets were resuspended in their respective lysis buffers (see below). Resuspended cells were disrupted by sonication. After centrifugation (30 min, 40,000$g$), the clear supernatant was filtered (0.45 $\mu$m; Millipore Corporation). The filtered supernatant was mixed with 2 ml slurry of previously washed and equilibrated Glutathione Sepharose 4B beads (GenScript). After incubation, beads were washed with 100 volumes of their respective wash buffer (see below). SMPC2A GST–tagged protein had an additional step, where beads were washed with 135 ml of wash buffer 2 (see below). An overnight cleavage of the GST-tag of all constructs was performed by addition of Precission protease to the bead solution. The post-cleavage proteins were eluted with their respective wash buffers. GST-tagged SYT1C2B which was eluted with wash buffer plus 20 mM reduced Glutathione and was not cleaved. A final purification step was performed by using a HiLoad 16/60 Superdex 200 preparative grade column (GE Healthcare) equilibrated in their respective Gel Filtration (GF) buffer (see below). The size and purity of the recombinant proteins was verified by SDS–PAGE. SYT1C2A-, C2AB-, and SMPC2A-purified proteins were concentrated to 10 mg/ml for protein crystallization and SASX trials and were stored at –80°C.

C2AB lysis buffer had 50 mM Tris–HCl, pH 8.0, 500 mM NaCl, 10 mM EDTA, and 2 mM DTT, 0.01 mg/ml DNase, 0.1 mM PMSF, and 0.5× of EDTA-free Protease Inhibitor Cocktail (cOmplete; Roche). Wash buffer had 50 mM Tris–HCl, pH 8.0, 500 mM NaCl, and 2 mM DTT. GF buffer had 20 mM Tris–HCl, pH 8.0, and 50 mM NaCl.

SYT1C2A and SYT1C2A-PolyB lysis buffer had 50 mM Tris–HCl, pH 8.0, 500 mM NaCl, 1 mM CaCl$_2$, 2 mM DTT, 0.01 mg/ml DNase, and 0.1 mM PMSF. Wash buffer had 50 mM Tris–HCl, pH 8.0, 500 mM NaCl, 1 mM CaCl$_2$, and 2 mM DTT. GF buffer had 50 mM Tris–HCl, pH 8.0, 50 mM NaCl, and 1 mM CaCl$_2$. For GST-tagged SYT1C2B and SYT1C2A-DADA, the buffers used were the same without CaCl$_2$.

SMPC2A lysis buffer had 20 mM Tris–HCl, pH 8.0, 250 mM NaCl, 5% glycerol, 10 mM EDTA, 0.2% Triton X-100, 2 mM DTT, 0.01 mg/ml DNase, 0.1 mM PMSF, and 0.5× of EDTA-free Protease Inhibitor Cocktail (cOmplete; Roche). Wash 1 buffer had 20 mM Tris–HCl, pH 8.0, 250 mM NaCl, 5% glycerol, 10 mM EDTA, 0.2% Triton X-100, and 2 mM DTT. Wash 2 buffer and GF buffer were the same and had 20 mM Tris–HCl, pH 8.0, 100 mM NaCl, 5% glycerol, and 2 mM DTT.

### Crystallization, data collection, structure solution, and refinement

C2A with 1 mM CaCl$_2$ from purification buffer was concentrated to 10 mg/ml. Rhomboidal crystals appeared in two crystallization conditions one containing 20% wt/vol polyethylene glycol 6,000, 100 mM Tris, pH 8.0, and 10 mM zinc chloride and another one containing 12% wt/vol polyethylene glycol 3,350, 100 mM Hepes, pH 7.5, 5 mM cobalt (II) chloride hexahydrate, 5 mM nickel (II) chloride hexahydrate, 5 mM cadmium chloride hydrate, and 5 mM magnesium chloride hexahydrate. Crystals appeared in 2–7 d. Crystals were cryoprotected in the crystallization solution containing 25% wt/vol ethylene glycol, mounted in fiber loops and flash-cooled in liquid nitrogen. X-ray diffraction data were collected at the BL13-Xaloc, ALBA synchrotron using a Dectris Pilatus 6M detector.

Diffraction data were processed and scaled using XDS (Kabsch, 2010). AIMLESS from the CCP4 package (Collaborative Computational Project, Number 4, 1994) was used to merge the data (Winn et al, 2011). A summary of the data collection statistics is given in Table S1. The X-ray structures of C2A in complex with Ca$^{2+}$ and Cd$^{2+}$ were solved by molecular replacement with the program Phaser (Adams et al, 2010) using the coordinates of the C2A domain of Synaptotagmin-7 (Protein Data Bank [PDB] ID code 6ANK). Successive cycles of automatic refinement and manual building were carried out with Phenix Refine (Adams et al, 2010) and Coot (Emsley & Cowtan, 2004) Model stereochemistry was verified with Mol-Probity (Zhang et al, 2010). The protein cartoon figures were obtained using PyMOL (DeLano, 2002). The refinement statistics are summarized in Table S1.

### Small-angle X-ray scattering (SAXS)

SAXS data were collected in the bioSAXS beamline B21, at Diamond Light Source, Harwell, United Kingdom (Cowieson et al, 2020). For the SYT1C2AB and SYT1SMPC2A 45 $\mu$l for each with a concentration of 10 and 6.6 mg ml$^{-1}$, respectively, were injected onto a Superdex 200 increase 3.2/300 column at 20°C using 50 mM Tris, pH 8, 50 mM NaCl and 20 mM Tris, pH 8, 100 mM NaCl, 5% glycerol, 2 mM DTT, respectively, as the running buffer. The output flow from the Agilent HPLC was directed through a 1.6 mm diameter quartz capillary cell held in vacuum. The flow rate was set to 0.08 ml min$^{-1}$ and 620 frames (with an exposure time of 3 s) were collected using a PILATUS 2M (Dectris) detector at the distance of 3.7 and 4.014 m from SYT1C2AB and SYT1SMPC2A samples, respectively. Collected two-dimensional images were corrected for variations in beam current, normalized for exposure time and processed into one-dimensional scattering curves using GDA and the DAWN software (Diamond Light Source). The background was manually subtracted using the program ScÅtter (https://www.bioisis.net/tutorials/9). SAXS data

collection and experimental parameters are summarized in Table S2.

Analysis of the one-dimensional SAXS experimental curves was initially performed to judge the quality of the data and obtain basic structural information related to the size and shape of the studied proteins (Fig S4 and Table S2). One such structural parameter is the radius of gyration ($R_g$) calculated from the slope of the Guinier plot described as $\ln(Iq)$ versus $q^2$, where $q = 4\pi\sin(\theta)/\lambda$ is the scattering vector ($2\theta$ is the scattering angle and $\lambda$ is the wavelength) (Guinier, 1939). For globular proteins, this plot is expected to be linear at low $q$, corresponding to values of $q \times R_g < 1.3$ in the Guinier zone of $0-1/R_g$. Linearity of the Guinier plot is considered a quality measurement of the data but does not ensure ideality of the sample. SAXS curve analysis also provides an estimate on the molecular mass of a protein that in turn relates to its oligomeric state. The molecular mass of a protein approximately corresponds to half of its Porod volume (excluded volume of the hydrated protein particle). Porod analysis reflects the behavior of the scattering intensity at higher $q$ range (Porod plot). Finally, the maximum size of a protein ($D_{max}$) can be obtained from analysis of the SAXS data by means of the pair-distance distribution function [P(r)], which corresponds to the distribution of distances between all the electrons within the protein. The pair-distance distribution function is obtained using the indirect Fourier transformation (Glatter, 1977), with a trial and error procedure at the end of which the obtained $D_{max}$ corresponds to the smoothest and positive distribution. Differences in the $D_{max}$ of a protein relate to conformational changes. In addition, it is possible to calculate the $R_g$ from the pair-distance distribution function and compare its value with that estimated from the Guinier plot. SAXS parameters for all the proteins are listed in Table S2.

The $R_g$ values calculated using the P(r) function for C2AB and for SYT1C2A are in good agreement with those estimated from the Guinier plot. The Porod volume and Porod exponent were used to estimate the molecular weight and the flexibility. The dimensionless Kratky plot was used to investigate the flexibility (and shape) of the proteins (Fig S5). The maximum value of 1.104 at $\sqrt{3}$ (dashed black line) corresponds to a globular and compact protein, similar to the BSA protein used as a standard in these experiments. In all of our cases the maxima are shifted to the right (larger than $\sqrt{3}$) and the maximum is higher than the standard, which denotes a well-folder but asymmetric shape for them.

## Molecular docking

PS and PI(4,5)P$_2$ were used for virtual screen against SYT1C2A structure in complex with Ca$^{2+}$ using GOLD 5.4.1 (Genetic Optimisation for Ligand Docking) software (Jones et al, 1997). Ligand docking was performed using the fast genetic algorithm search option; the protein was treated as rigid and full flexibility was allowed for the ligands; the binding site was defined as the residues with at least one heavy atom within 10 Å from the centroid of the residues in the basic crevasse where the reference ligands are placed (PDB codes 3GPE, 4NP9, and 4NS0); early termination was allowed if the three top solutions are within 0.5 Å. The GOLD software (using the default CHEMPLP scoring function) automatically scores the resulting ligand solutions and it selects those complexes that represent potentially meaningful interactions.

## Molecular modeling and MD simulations

The CHARMM-GUI Membrane Builder pipeline (Wu et al, 2014) was used to construct two model systems consisting of SYT1C2A in juxtaposition to the inner leaflet of two distinct well-packed lipid bilayers, PS-M and PSPIP-M. The PI(4,5)P$_2$-free PS-M membrane was composed of 152 dimyristoylphosphoserine (DMPS) and 18 sitosterol molecules in the outer layer and 145 DMPS and 18 sitosterol molecules in the inner layer; the PI(4,5)P2-enriched PSPIP-M membrane contained 158 DMPS and 20 sitosterol molecules in the outer layer and 45 DMPS, 90 dimyristoylphosphatidylinositol(4,5) bisphosphate (DMPI), and 18 sitosterol molecules in the inner layer. The use of sitosterol or stigmasterol in research involving plant membranes and membrane-associated proteins is recommended (Zhuang et al, 2017) to recapitulate the fluidity and thermotropic properties of model membranes (Jovanović et al, 2019). Sitosterol, as is the case for cholesterol in mammal membranes, contributes to packing the lipid bilayer more tightly into a liquid ordered phase (Emami et al, 2017), and this aspect is important in MD simulation to attain equilibration more effectively (Silva et al, 2011). Myristic acid was chosen as a representative fatty acid for computational tractability. The initially identical protein orientation in both cases was unbiasedly dictated by the PPM web server that makes extensive use of the Orientations of Proteins in Membranes database (Lomize et al, 2012). The whole protein and the lipid head groups on both sides of the membrane were then solvated along the z axis with a 10-Å thick layer of TIP3P water molecules (~26,500 in all) and the bulk ion concentration was set at 0.15 M KCl. The AMBER *ff14SB* (Maier et al, 2015) and *lipid14* (Dickson et al, 2014) force fields were used for protein and lipids, respectively. Consistent parameters and point charges were derived for sitosterol (Madej et al, 2015) and DMPI using *antechamber* in AMBER18 (https://ambermd.org/).

The geometry of both complexes was optimized by carrying out a series of progressive energy minimizations, essentially as described (Perona et al, 2020), and the resulting coordinate sets were first heated up to 100°K over 5 ps at constant volume using the Langevin thermostat and thereafter to 303°K over 100 ps. This temperature was maintained during 5 ns to equilibrate the system's dimensions and density at a constant pressure of 1 atm by means of a Berendsen barostat and anisotropic scaling. The unrestrained MD simulations under these conditions were further extended for data collection using the *pmemd.cuda* code (Salomon-Ferrer et al, 2013), as implemented in AMBER18, running on single Nvidia graphics procesing units up to a total simulation time of 485 ns. The coupling constants for the temperature and pressure baths were 1.0 and 0.2 ps, respectively. The application of SHAKE to all bonds allowed an integration time step of 2 fs to be used. The cutoff distance for the nonbonded interactions was 9 Å and the list of nonbonded pairs was updated every 25 steps. Periodic boundary conditions were applied and electrostatic interactions were represented using the smooth particle-mesh Ewald method (Darden et al, 1993) with a grid spacing of 1 Å. Trajectory snapshots were saved every 0.1 ns and figures generated by PyMOL were concatenated for movie production using ImageMagick (https://imagemagick.org).

All-atom structural models of SYT1C2B, SYT1C2AB, and SMPC2A were built using the threading methods implemented in the Phyre 2.0 (Kelley et al, 2015) and Swiss-Model (Biasini et al, 2014) servers by providing the amino acid sequence deposited in UniProtKB under the code Q9SKR2 (SYT1_ARATH). The best protein template of known 3D structure was PDB entry 4P42 (DOI: 10.1038/nature13269). The RCD+ server (López-Blanco et al, 2016) was used to fill in existing gaps and generate alternative loop conformations of the intrinsically disordered regions. Membrane insertion, energy minimization, and unrestrained MD simulations proceeded as explained above for SYT1C2A. All the simulations revealed a stable association of SYT1 with membranes (Fig S4F and H).

### ITC

ITC experiments were carried out at 25°C using a MicroCal VP-ITC (GE-Healthcare). The C2A construct was equilibrated in 50 mM Hepes, pH 8.0, and 50 mM NaCl; $Ca^{2+}$ solutions were prepared using the same buffer. Samples were degassed before use. Titration was carried out by injecting consecutive aliquots of 20 mM $Ca^{2+}$ ($1 \times 1$ $\mu l$, $10 \times 5$ $\mu l$, and $10 \times 10$ $\mu l$) into the sample cell loaded with 151 $\mu M$ C2A. Heat that developed on $Ca^{2+}$ dilution was found to be negligible. The thermodynamic parameters of binding were calculated analyzing the binding isotherm with the MicroCal ITC Origin software. Three independent measurements were carried out. The experimental conditions were the same except for the ligand protein ratio. We sampled 1:10, 1:14, and 1:16 obtaining similar results that indicate a single high affinity $Ca^{2+}$ binding site ($K_d$ = 0.06 ± 0.02, 0.0001 ± 0.02 and 0.0001 ± 0.02 $\mu M$; number of sites N = 0.221 ± 0.003, 0.359 ± 0.002 and 0.301 ± 0.001 and ΔH = −5.3 ± 0.1, −5.4 ± 0.1 and −5.6 ± 0.1 kcal•$mol^{−1}$).

### Thermal shift assay

Label-free thermal shift assays with SYT1C2A, SYT1C2A-DADA, SYT1C2A-PolyB, SYT1C2B, and GST as negative control were performed using Tycho NT. 6 (NanoTemper Technologies). Assays were performed at 5.8 $\mu M$ protein in 0.2 mM Tris–HCl, pH 8.0, 0.5 mM NaCl and run in duplicates in Tycho NT.6 capillaries (Cat no. TY-C001; NanoTemper Technologies). Different protein solutions were prepared to reach a final $CaCl_2$ concentration of 0.01, 0.03, 0.3, and 3 mM. Ligand binding assays were performed at 58 $\mu M$ ligand and at 0.02 mM EGTA. Because SYTC2B data were not completely sigmoidal, a smoothing of the first derivative data was performed every 1°C for $Ti$ calculation. Intrinsic fluorescence was recorded at 330 and 350 nm while heating the sample from 35°C to 95°C at a rate of 30°C/min. The ratio of fluorescence (350/330 nm) and the $Ti$ were calculated by Tycho NT. 6.

### BLI

Lipid-protein interactions were measured by BLI using a single channel BLItz system (ForteBio). Small Unilamellar Vesicles of PC/PS (phosphatidylcholine/phosphatidylserine) 75:25 and PC/PS/PI (phosphatidylinositol) 65:25:10 stocks in chloroform at 2 mg/ml were diluted in SLB buffer (50 mM Tris–HCl, pH 7.5, 150 mM KCl)

to a final concentration of 0.5 mg/ml, vortexed for 5 min, and sonicated for 15 min until a homogeneous solution was achieved. Using this solution, lipid monolayers were formed and immobilized on a previously hydrated aminopropylsilane biosensor tip (ForteBio). All the protein fragments at 0.5 $\mu M$ were posteriorly associated to the Small Unilamellar Vesicles in 50 mM Tris–HCl, pH 8.0, 50 mM NaCl, and 10 $\mu M$ $CaCl_2$ or 100 $\mu M$ $CaCl_2$ for PC/PS or PC/PS/PI, respectively. $Ca^{2+}$-free assays were performed with 20 $\mu M$ EGTA. $IP_3$-binding assays were performed at 5.8 $\mu M$. In the case of SYT1C2a-PolyB, BSA was previously associated to discard unspecific interactions with the lipid vesicles. A positive control associating BSA before SYT1C2A was also carried out. For SYT1C2B, a subtraction of the sensogram corresponding to the unspecific binding of GST was performed. Binding reactions consisted of a 60 s baseline (buffer), followed by a 300-s association phase (membranes or protein binding), and a 330 s dissociation phase (buffer only). BLItz Pro software was used to analyze experimental results.

### Phospholipid-binding assays

Phospholipid mixtures of PC/PS 75:25 and PC/PS/PI 65:25:10 and pure PC in chloroform were dried under a stream of nitrogen until a thin layer of lipids was obtained. Dried phospholipids were kept under vacuum overnight and were posteriorly resuspended in Liposome Buffer (50 mM Tris–HCl, pH 7.5, 80 mM KCl, and 5 mM NaCl) to a final concentration of 1 mg/ml. Lipids were left in Liposome Buffer for 25 min at 20°C before sonicating for 7 min. Afterwards, the liposomes were passed 11 times through 0.8 $\mu m$ Nucleopore polycarbonate filters (Whatman) by syringe extrusion. Protein-liposome mixtures (40–100 $\mu g$) were incubated for 30 min at 25°C in a final volume of 200 $\mu l$. Assays were performed at a final $Ca^{2+}$ concentration of 300 $\mu M$. $Ca^{2+}$-free assays were performed with 20 $\mu M$ EGTA. Controls of all samples were prepared by the addition of Liposome Buffer instead of phospholipids. GST binding to liposomes was also measured as negative control. Samples were centrifugated at 58,000 rpm using a TLA100 rotor (Beckman) for 1 h at 4°C to pellet the liposomes. To effectively measure the amount of protein present in the supernatant, the intrinsic fluorescence of the unfolded samples was measured at 80°C at 330 and 350 nm to subtract the effect of the chemical environment on the tryptophan using Tycho NT. 6. The amount of protein seized by the phospholipids was calculated as the percentage of the difference between the fluorescence of the control and the sample. Controls were considered to be the total amount of protein present in the assays.

## Data Availability

The atomic coordinates and structure factors data from this publication have been deposited in the PDB, https://www.pdb.org/ (PDB ID codes for $Ca^{2+}$ and $Cd^{2+}$ complexes of SYT1C2A [7ATP and 7AS6, respectively]). The final SAXS models were deposited and are available at SASBDB https://www.sasbdb.org/ (ID codes SASDKG6 for the SMP2C2A construct SASDKJ9 for the C2AB construct and SASDKK9 in presence of $Ca^{2+}$).

## Supplementary Information

## Acknowledgements

This work was funded by grants from Agencia Estatal de Investigación (AEI, Spain) and Fondo Europeo de Desarrollo Regional (European Union) (BIO2017-89523-R to A Albert). A Albert and D Siliqi thank ALBA (beamline XALOC) and Diamond Light Source (beamline B21, proposal MX21741) for granting access to the synchrotron radiation source. We thank Nathan Cowieson and Charlotte JC Edwards-Gayle from Diamond Light Source for their assistance in the preparation of the SAXS experiments.

### Author Contributions

JL Benavente: formal analysis, investigation, and writing—original draft, review, and editing.
D Siliqi: formal analysis, investigation, and writing—review and editing.
L Infantes: formal analysis and investigation.
L Lagartera: data curation and investigation.
A Mills: investigation.
F Gago: formal analysis, investigation, and writing—review and editing.
N Ruiz-López: conceptualization and writing—review and editing.
MA Botella: conceptualization and writing—review and editing.
MJ Sánchez-Barrena: conceptualization and writing—review and editing.
A Albert: conceptualization, formal analysis, supervision, funding acquisition, investigation, project administration, and writing—original draft, review, and editing.

### Conflict of Interest Statement

The authors declare that they have no conflict of interest.

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
