## [Reviewer comments · Life Science Alliance]

Life Science Alliance

The structure of the Arabidopsis Synaptotagmin 1 reveals its regulation at membrane contact sites

Juan Benavente, Dritan Siliqi, Lourdes Infantes, Laura Lagartera, Alberto Mills, Federico Gago, Noemi Ruiz-Lopez, Miguel Botella, Maria Sanchez-Barrena, and Armando Albert

DOI: <https://doi.org/10.26508/lsa.202101152>

Corresponding author(s): Armando Albert, Consejo Superior de Investigaciones Cientificas

Review Timeline:	Submission Date:	2021-07-08
	Editorial Decision:	2021-07-12
	Revision Received:	2021-07-21
	Accepted:	2021-07-21

Transaction Report:

Please note that the manuscript was reviewed at Review Commons and these reports were taken into account in the decision-making process at Life Science Alliance.

Review
COMMONS

1st Authors' Response to Reviewers

Reviewer 1

Major points

The conclusion that SYT1C2A determines protein behavior by binding calcium and likely switching its function from a pure tether to a lipid transporter is accurately and convincingly portrayed. However, there are some problematic statements in the way this information is interpreted and used to propose a model.

- *p7: "This means that site 1 will be occupied depending on the physiological Ca²⁺ concentration to activate the protein and trigger Ca²⁺-dependent SYT1C2A lipid binding." Please exchange "means" by "suggests"*

We agree with the reviewer and we have exchanged "means" by "suggests"

- *There is not enough data to determine whether SYT1C2A interacts with the SYT1SMP domain, much less to propose that it is the SYTC2A that loads the SMP with DAG. They may also operate independently by means of the unstructured hinge sequence between them.*
 - *It is implied that SYTC2A would have to release its calcium atom for it to leave the PM and go to the ER to complete a single transport cycle. It is unclear whether SYT1C2A could bind and release calcium repeatedly and fast enough to cycle between membranes efficiently. It appears more*
-

plausible for it to remain at the PM until intracellular calcium concentrations decrease.

These statements should be marked as speculative.

We understand that the reviewer comments are related to steps 3 and 4 of our model for the function of SYT1 (figure7; pages 17 and 18). We have now included these scenarios suggested by the reviewer (page 17 and 18). In addition, we have discussed the work from Vennekate et al, PNAS, 2012, related to the cis and trans membrane interactions of the two C2 domains of human synaptotagmin (Syt1). The authors demonstrated that they are driven by the balance in the concentration of anionic lipids between target membranes and by Ca^{2+} and protein concentration. Likewise, the changes in the local composition of PIP and/or Ca^{2+} may drive the transfer of SYT1C2A from the PM to de ER and *vice versa* while the C2B remains at the PM.

Most of the graphs shown (except Fig. 4A) appear to show only one experiment. Please include results from at least three independent experiments in all graphs, and analyze them statistically. Please describe only differences between samples that are statistically significant (for all graphs and tables).

Following reviewed recommendation, we have assured that all the biochemical data arises from three independent measurements. Indeed, as indicated in the figure legends and methods section, this was the case for the Ca^{2+} binding experiments at figure 2C and for the lipid binding data at figure 4. We have now repeated the ITC experiments to provide three independent measurements of the dissociation constant of SYT1C2A for Ca^{2+} .

Minor comments

Please include the PDB codes for the available structures used. It is not clear where the C2C domain structure comes from. We guess it may correspond to PDB entry 2DMG. A word of caution, this 2006 structure has been released but not published and its quality is suboptimal. After its appearance in Idevall-Hagren et al. 2015 supplementary images it has been re-cited in other publications. I would advise against making topology claims based on this structure (Figure S1), and only use it as an example for polybasic patches in C2 domains.

We have included the PDB codes for the available structures used at the corresponding figure captions 1, 3 and S1.

We agree that the stereochemistry of the refined structure of the E-Syt2 C2C domains (2DMG) is suboptimal. However, we have sought for the advice of an NMR expert (M.A. Jimenez, <https://scholar.google.es/citations?hl=es&user=iAF0ymYAAAAJ>) to evaluate the NMR restrains available alongside the entry at PDB. She concluded that the Halpha-Halphi distance restrains indicate that the connectivity of the beta strands is correct. The confusion may arise from the fact that the PDB entry has been updated twice since 2006, the last one in 2011.

- p4: Schauder et al. 2014. Only the "shuttle" model is mentioned in the text. Although this may be the most plausible scenario, the "tunnel" model cannot be completely ruled out.

We have now stated that both shuttle or tunnel modes are possible for lipid transfer at page 4.

- p5: Fernandez-Busnadiego et al. 2015 does not make a reference to the claim in the text.

Fernandez-Busnadiego et al. 2015 has been replaced by Collado et. al 2019

- p18: Collado et. al 2019 don't assign the peak-forming function to the N-terminal hairpin domain of tricalbins.

As the reviewer indicates, Collado et. al 2019 just suggests that the peak forming function could be due to the N-terminal hairpin domain of tricalbins: *"This phenomenon may rely on the hairpin sequence that anchors Tcbs to the ER membrane. Tcb hairpin sequences could sense and/or generate membrane curvature as in reticulons and other ER morphogenetic proteins (Hu et al., 2011)."* (Taken from Collado et al. 2019).

Accordingly, we have indicated that "It has been suggested that the insertion of the N-terminal hydrophobic end of Tcbs may induce the formation of peaks of strong curvature at the ER region facing the PM. These structures shorten the distance between the ER and the PM by ~7 nm and facilitate lipid transport (Collado et al., 2019)"

Fig. 1A: the notation for the domains C2C-C2E of E-Syt1 is confusing and not described in the legend. Also, please mark which beta sheet is which more clearly in Fig 1B, it is very difficult to understand the labelling Fig. 1C: several elements (e.g. types of boxes, "T" on the top) used in the figure are not described in the legend. Both beta sheets and mutations are described as "arrows" in the legend, please differentiate between vertical and horizontal.

We have modified the Fig. 1A legend to explain the notation of C2 domains of human E-Syts "Human E-Syt1 displays five C2 domains while E-Syt2 and E-Syt3 display three".

Labeling of beta strands has been modified to make the panel clearer.

We have rewritten the caption of figure 1C to describe the types of boxes and the "T" and to distinguish between vertical and horizontal arrows.

Fig. 2A: indicate Lys 286, which is mentioned in the text

The reference to Lys 286 in the text was not correct. We have modified the text to indicate that Lys275 replaces the Ca I as it is shown in figure 2.

Fig. 2B: Inset too small to read.

Labels for inset at Fig 2B have been enlarged.

Fig. 3: please show the scale and explain the blue-red color code. Please mark the polybasic patch.

We have explained the meaning of the blue and red code, marked the polybasic patch of SYT1C2A and E-Syt1C2C and indicated that all of them are scaled to the same value.

Fig. 4A should show both mutants for both types of liposomes.

We have included new liposome binding data including both mutants for both types of liposomes.

Table 1: what is "control"?

We have indicated that the unbound wild-type protein is taken as a control

Fig. S5: Difficult to read, increase font size. The text talks about experiments with calcium and EGTA that are not shown in the figure.

We have increased figure size to facilitate reading. We have also indicated that C2AB-Ca and C2A stand for the experiments carried out in presence of Ca and EGTA, respectively.

Reviewer 2

Minor comments

Typos: please re-read carefully through the manuscript to remove them. We advise the authors to have the manuscript corrected by a native english-speaker.

We have done a thorough revision of the manuscript to correct typos and to improve the English style of the manuscript

Reviewer 2 considers that "The authors discuss their findings within the frame of experimental observations that are already published but these remain speculative"

We are glad that the reviewer appreciates our work in "decrypting the roles of C2 individual operating modes" as it is a "central issue for providing functional specificity but also plasticity in response to developmental /environmental clues". The reviewer also considers our work "important and identifies a number of very interesting features". As already mentioned to the editor, the present version of the manuscript includes new biochemical data and analysis to support further the functions of C2A-C2B tandem. In addition, we have included new references and rephrased some of the headings and statements in the discussion section, to adjust them better to our experimental results.

Reviewer 3

Major points

1)The authors investigated SYT1C2A calcium-binding sites using two different methods, ITC and differential scanning fluorimetry. By using ITC, they described the first binding site coordinating calcium ions in the nanomolar range. The second calcium-binding site was then characterized by differential scanning fluorimetry. The second calcium-binding site binds calcium with Kd of 277 μ M. The authors then mutated SYT1C2A at two positions and performed again differential scanning fluorimetry. In this case, they did not observe any blue shift in intrinsic fluorescence concluding that "calcium-binding is mediated by the calcium-binding site". It is not clear which binding site the authors mean. In the structure of SYT1C2A, the mutated residues (D276 and D282) are shared by both calcium-binding sites. It is, therefore, difficult to interpret the data. The authors should generate a unique mutation for each site and perform both ITC and differential scanning fluorimetry

The SYT1C2A-DADA double mutant was prepared to abolish Ca²⁺ binding to both site I and site II and to discard an unspecific effect of Ca²⁺ on intrinsic fluorescence that accounts for the low affinity Kd. Our data showed that the addition of Ca²⁺ to SYT1C2A-DADA does not produce a shift in intrinsic fluorescence of the protein; indicating that the observed Ca²⁺-binding activity is specifically mediated by the Ca²⁺ dependent lipid binding site. We clarify this point in the present version of the manuscript.

In addition, following reviewer's recommendation, we have prepared two additional point mutant proteins, SYT1C2A D276A and SYT1C2A E340A, to investigate the Ca²⁺ binding properties at site II and I, respectively. As expected, the reduction of one carboxylate ligand at the structural site II produces a drastic decrease in the Ca²⁺ binding affinity (Kd = 1.8 \pm 0.5 mM) which is coupled with a decrease in thermal stability of the protein (Ti = 55°C) and a red-shift in fluorescence emission with respect to the wild type protein that resembles those effects observed for the SYT1C2A-DADA mutant (Figure S2A). Differentially, SYT1C2A Glu334Ala doubled the Kd (534 \pm 60 μ M) while reducing slightly its thermal stability (page 7 figure S2A)

2)The authors described an increase in the inflexion point temperature with increasing calcium concentration. They noted that the effect was measurable from 30 μ M to 300 μ M. Looking at the plot with the first derivative ratio (Figure 2C), there is also an apparent change in the inflexion point temperature from 300 μ M to 3 mM. Does this mean that the SYT1C2A domain binds more than two calcium ions?

The ligand induced stabilization of proteins results in changes of the thermally induced melting curves for the ligand complexed relative to the uncomplexed proteins. This effect is used to unequivocally identify ligand hits for a particular protein from large libraries of compounds and to provide an initial estimation of the binding affinity. However, a deeper analysis of the ligand binding affinities using this technique is discouraged as the increase of melting temperature with the ligand concentration does not saturate to a particular value. This is why the effect of Ca²⁺ addition to SYT1C2A was also measurable from 300 μ M to 3 mM, and it does not necessarily imply the binding

of Ca²⁺ to another site. Consequently, we used other techniques such as ITC or the analysis of the change of intrinsic fluorescence upon Ca²⁺ addition to precisely characterize the Ca²⁺ binding affinities of SYT1C2A. In the present version of the manuscript, we have indicated that the change in Ti vs Ca²⁺ concentration is measurable when moving from 30 μM to 300 μM, thus demonstrating a Ca²⁺-binding event “is initiated” in this concentration range (page 7)

3) To address lipid-binding properties of the SYT1 C2 domains, the authors used two independent methods, lipid co-sedimentation assay and BLI. In the Material and Methods section, the authors wrote that a solution of liposomes was sonicated for seven minutes to achieve homogeneity. How was homogeneity checked? Standard protocols for the liposome co-sedimentation assay use the extruder to achieve a homogeneous population of the liposomes. Also, the authors noted that the liposomes were resuspended in the buffer containing 50 mM Tris/HCl, 80 mM KCl, and 5 mM NaCl. The liposomes are usually loaded with sugar molecules, like raffinose at this step to allow subsequent co-sedimentation using centrifugation.

Following the reviewer recommendation, we have used an extruder to achieve a homogeneous population of the liposomes (see the Methods section). This has produced an improvement of the data from the statistical point of view. However, we did not employ any sugar to facilitate lipid sedimentation as we found that 1 hour centrifugation at 58,000 rpm using a TLA100 rotor (Beckman) was enough to separate adequately soluble and precipitated fractions.

The authors wrote that the samples were centrifuged at 58,000 rpm. The information does not allow reproducibility without rotor specification.

We have now included the rotor specification in the methods section.

The bound protein was estimated by subtracting the supernatant from the total amount of protein used in the assay. Direct estimation of the protein amounts in the bound fraction via e.g. SDS-PAGE would be more suitable.

We respectfully disagree with the reviewer in this issue. The reviewer may note that the differences in the fraction of bound protein to the liposomes are at maximum around 25%. Such values are statistically significant using spectrophotometric techniques but there will be less accurately determined by the analysis of an SDS-PAGE. In addition, the later would require several washing steps of the insoluble fraction that may induce additional errors. This is well documented in a previous work from our group (Diaz et al, PNAS 2015). There, the lipid binding properties of WT and mutant C2 domain are compared using both approaches; in this work it is shown that the spectrophotometric techniques showed significant differences with the SDS-PAGE, which resulted just indicative.

Nevertheless, we have prepared the requested SDS-PAGE for reviewer evaluation (see below) and if required we will include it as a new panel in figure 4 or include it as supplementary material. The SDS-PAGE shows the amount of soluble protein after the incubation with liposomes. It is clearly

shown a reduction in the amount of WT and DADA soluble protein upon incubation with PCPSPI liposomes with respect to the sample incubated with PCPS liposomes. Differentially, no effect is observed for the PolyB and WT in presence of EGTA. Sample migrates abnormally producing a double band in presence of EGTA, probably due to the effect of removing the structural Ca site.

Critical controls for the liposome co-sedimentation assay are missing:

Do the SYT1 C2 domains bind liposomes without negatively charged lipids (i.e. PC-only liposomes)?

Does the SYT1C2A-DADA mutant domain bind the PC/PS/PI liposomes?

Does the SYT1C2A-PolyB mutant interact with the PC/PS liposomes?

We have included all the controls suggested by the referee in the present version of the manuscript, in the results section and in figure 4A.

The authors wrote both in the main text and the figure 4 legend that they used a lipid monolayer in the BLI method. However, in the Material and Methods section, they wrote that they used small unilamellar vesicles.

The starting material for lipid monolayer immobilization at the biosensor tip is a solution of small unilamellar vesicles. We have clarified this issue in the methods section.

4)The authors beneficially used all-atom MD simulations to address mechanistic details of the SYT1 C2 domains with two different lipid bilayers. However, several issues need to be addressed.

Is there a particular reason to include sitosterol in the MD simulations? Does sitosterol contribute to protein binding?

We clarify this issue in the present version of the manuscript. The use of sitosterol or stigmasterol in research involving plant membranes and membrane-associated proteins is recommended (DOI: 10.1063/1.4983655 PMID: 28595398) to recapitulate the fluidity and thermotropic properties of model membranes (DOI: 10.1016/j.colsurfb.2019.110422 PMID: 31437609). Sitosterol, as is the case for cholesterol in mammal membranes, contributes to packing the lipid bilayer more tightly into a liquid ordered phase (DOI: 10.1016/j.chemphyslip.2017.01.003 PMID: 28088325) and this

aspect is important in molecular dynamics simulations to attain equilibration more effectively (DOI: 10.1016/j.jcis.2011.02.048 PMID: 21429500).

We observed one hydrogen bond contact between Asn 338 at loop L3 and sitosterol in PSPI-M. We clarify this point in the results section.

Why was not the simulation ran for a longer time? What was a criterion to determine that the system reached a stable state? Results of the MD simulations are presented as static snapshots. The manuscript would benefit from a more detailed analysis, e.g. the number of hydrogen bonds between the protein and phospholipid molecules over time, development of the tilt angle over time, etc.

Following the reviewer's recommendations, we now provide data showing the dynamic features of our MD calculations. In particular, we have compared the overlay of the different structures along the simulation of the SYT1C2A domain attached to the PS-Membrane and to the PSPIP-Membrane highlighting those amino acids hydrogen bonded to phospholipids (Figure S4A and S4B). This picture illustrates well that the pattern of interactions is conserved along the simulation. In addition, it also shows that the orientation of the domain with respect to the plane of the membrane is conserved. In this respect, as the reviewer requested, we have also included, a picture illustrating the change in the tilt angle of the C2A with respect to the plane of the membrane along the simulation. This analysis reveals that the tilt angle with respect to the membrane is 20 degrees larger for the PSPIP-Membrane than for the PS membrane due to the interaction of PIP molecules with the polybasic site (Figure S4E).

In addition, we now present time traces illustrating the course of the simulations. In particular, those corresponding to the RMSD of the individual SYT1C2A, SYT1C2B and the linker between them. They clearly show that the simulations reached equilibrium within the time sampled (Figure S5H).

The authors performed the MD simulations for both SYT1 C2 domains. It would be informative to include an electrostatic potential mapped on the surface of the SYT1C2B domain similarly to figure 3. The experimental results showed that SYT1C2B binds liposomes containing PC/PS. How does SYT1C2B interact with the PC/PS membrane? The authors noted that SYT1C2 inserts loop L3 into the lipid bilayer and that this loop adopts a β -hairpin structure. This is the case for the simulation with the SYT1C2AB fragment, but not for the SYT1C2B domain alone. Is the β -hairpin formed during the MD simulation? Or is it a result of the template-based modelling?

To clarify reviewers' concerns, we have included a supplementary figure illustrating the RMSD per residue along the MD simulation for the SYT1C2AB protein fragment in solution, and attached to PSPI-M. The representation illustrates an overall reduction of the RMSD as a result of the protein stabilization at the membrane, which is more significant at the membrane binding loops. In particular, the highly flexible SYT1C2B L3 loop in solution becomes highly stabilized upon membrane interaction and it folds as a beta hairpin.

We have calculated the electrostatic potential mapped on the surface of the SYT1C2B. As it does not come from an experimental structure, we decided not to include in figure 3. The map, which is

depicted below in the orientation shown in figure 3, shows that SYT1C2B might not display a polybasic site in accordance with our experimental results.

How is the proposed binding model of the SYT1C2AB affected by the data obtained using SAXS?

SAXS data showed that SYT1C2AB fragment may adopt a V-shaped compact structure and an extended conformation in which there is no interaction between the C2A and C2B domains. Interestingly, the V-shaped structure in solution resembles the proposed binding mode for SYT1C2AB to the membrane. We clarify this point in page 15.

Minor comments

1) SYT1CB construct is not listed in figure 1A along with the other constructs used in the study.

We have modified Figure 1B to include SYT1C2B

2) Phospholipids contain a phosphate group rather than a phosphoryl group.

We have modified the text to correct this.

3) In the text, the authors sometimes used the ratio 350 nm/330 nm and sometimes 330 nm/350 nm.

This has been corrected in Figure 2C

4) Figure S2, displayed curves look strikingly similar, different line representation does not allow a proper comparison. There are no units for y-axes.

The figure has been corrected according to the reviewer's recommendation for proper comparison.

5) Figure 4B, y-axes do not have scales.

The Y-axis from the BlitZ system represents the thickening of the layer of proteins attached to the biosensor tip and it is given in nm. However, this parameter could be meaningless when comparing different membranes and/or proteins. Hence, we scaled the plots to the maximum thickness for comparison purposes.

6) Figure 5B, IP3 or I3P?

We have corrected the labels corresponding to inositol triphosphate IP3 at figure 5B

7) The authors noted that based on their SAXS experiments, the SYT1-SMPC2A has a maximum size of 176 Å and they wrote that this value is in accordance with the size of the previously characterized E-Syt2-SMPC2AB protein. However, as the authors reported, E-Syt2-SMPC2AB is two times smaller.

We agree with the reviewer, the size of the *SMP tunnel of E-Syt2-SMPC2AB is two times smaller* than the Dmax size of SYT1 SMPC2A. We clarify this point in the text.

Yours sincerely

Armando Albert

July 12, 2021

RE: Life Science Alliance Manuscript #LSA-2021-01152-T

Dr. A Albert
Consejo Superior de Investigaciones Cientificas
Inst. Quimica Fisica Rocasolano
Serrano 119
Madrid, Madrid 28006
Spain

Dear Dr. Albert,

Thank you for submitting your revised manuscript entitled "The structure and flexibility analysis of the Arabidopsis Synaptotagmin 1 reveal the basis of its regulation at membrane contact sites". We would be happy to publish your paper in Life Science Alliance pending final revisions necessary to meet our formatting guidelines.

- please add ORCID IDs for Lourdes Infantes, Alberto Mills and Maria Jose Sanchez-Barrena
- please provide all figures (including Supplemental) as separate files
- please add a Movie legend in the manuscript (after the other supplemental figure legends)
- please add a Data Availability Statement if any structural information has been deposited somewhere

LSA now encourages authors to provide a 30-60 second video where the study is briefly explained. We will use these videos on social media to promote the published paper and the presenting author. Corresponding or first-authors are welcome to submit the video. Please submit only one video per manuscript. The video can be emailed to contact@life-science-alliance.org

A. FINAL FILES:

B. MANUSCRIPT ORGANIZATION AND FORMATTING:

Sincerely,

July 21, 2021

RE: Life Science Alliance Manuscript #LSA-2021-01152-TR

Dr. Armando Albert
Consejo Superior de Investigaciones Cientificas
Inst. Quimica Fisica Rocasolano
Serrano 119
Madrid, Madrid 28006
Spain

Dear Dr. Albert,

Thank you for submitting your Research Article entitled "The structure of the Arabidopsis Synaptotagmin 1 reveals its regulation at membrane contact sites". It is a pleasure to let you know that your manuscript is now accepted for publication in Life Science Alliance. Congratulations on this interesting work.

DISTRIBUTION OF MATERIALS:

Again, congratulations on a very nice paper. I hope you found the review process to be constructive and are pleased with how the manuscript was handled editorially. We look forward to future exciting submissions from your lab.

Sincerely,
